# Porous organic polycarbene nanotrap for efficient and selective gold stripping from electronic waste

Xinghao Li[1], Yong-Lei Wang[2], Jin Wen [1], Linlin Zheng[1], Cheng Qian [1], Zhonghua Cheng[1], Hongyu Zuo[1], Mingqing Yu[1], Jiayin Yuan [2], Rong Li[3], Weiyi Zhang [1] ✉ & Yaozu Liao [1] ✉

The role of N-heterocyclic carbene, a well-known reactive site, in chemical catalysis has long been studied. However, its unique binding and electron-donating properties have barely been explored in other research areas, such as metal capture. Herein, we report the design and preparation of a poly(ionic liquid)-derived porous organic polycarbene adsorbent with superior gold-capturing capability. With carbene sites in the porous network as the "nanotrap", it exhibits an ultrahigh gold recovery capacity of 2.09 g/g. In-depth exploration of a complex metal ion environment in an electronic waste-extraction solution indicates that the polycarbene adsorbent possesses a significant gold recovery efficiency of 99.8%. X-ray photoelectron spectroscopy along with nuclear magnetic resonance spectroscopy reveals that the high performance of the polycarbene adsorbent results from the formation of robust metal-carbene bonds plus the ability to reduce nearby gold ions into nanoparticles. Density functional theory calculations indicate that energetically favourable multinuclear Au binding enhances adsorption as clusters. Life cycle assessment and cost analysis indicate that the synthesis of polycarbene adsorbents has potential for application in industrial-scale productions. These results reveal the potential to apply carbene chemistry to materials science and highlight porous organic polycarbene as a promising new material for precious metal recovery.

Gold, well known for its unique physical and chemical properties, e.g., high electrical conductivity, excellent corrosion resistance and high melting point, is widely applied in jewellery, chemical catalysis, electronic devices, etc. Consequently, it is widely present in a broad range of scrap materials, e.g., in electronic wastes (e-wastes)[1,2]. The continuously growing consumption and the limited supply of gold, which is an irreplaceable high-value element, create a pressing need for increased production and recycling[3]. To date, the resourcing and recycling of gold, however, suffer from the low abundance of this metal in ore mines

and the inefficiency and high energy consumption of pyrometallurgy for ores and e-wastes, which involves many hazards that adversely induce secondary pollution and high cost[4]. As a consequence, the exploration of improved functional material-enabled selective and effective methods for "gold metallurgy" continues, particularly for urban mining of gold from e-wastes because the metal abundance of these materials is 10-100 times higher than that of natural ores.

In this context, approaches such as hydrometallurgy and biometallurgy have been developed for leaching gold from e-wastes.

[1]State Key Laboratory for Modification of Chemical Fibers and Polymer Materials, College of Materials Science and Engineering, Donghua University, Shanghai 201620, China. [2]Department of Materials and Environmental Chemistry, Stockholm University, Stockholm 10691, Sweden. [3]College of Chemistry and Chemical Engineering, Donghua University, Shanghai 201620, China. ✉e-mail: wyzhang@dhu.edu.cn; yzliao@dhu.edu.cn

A comparison of these methods shows that hydrometallurgy employs a more environmentally friendly and energy-efficient treatment by using digestive solutions to precipitate gold ions from e-waste, followed by selective gold ion extraction from the complex digestive solutions[5]. Conventional gold extraction pathways include solvent extraction[6], ion exchange[7], electrochemical reduction[8], and chemical/physical adsorption[9–13]. In these routes, chemical/physical adsorption is an environmentally benign option for gold recovery. Hence, a plethora of extraction materials, e.g., porous organic polymers (POPs), have been applied for the sustainable utilization of gold resources. These POPs include covalent organic frameworks (COFs), polymers of intrinsic microporosities (PIMs), porous aromatic frameworks (PAFs), and conjugated microporous polymers (CMPs). For instance, Qian et al.[14] prepared irreversible amide-linked COFs via a building block exchange strategy, which offered the fastest kinetics (reaching equilibrium in only 10 s) for the adsorption of $Au^{3+}$; Hong et al.[15] developed a family of porphyrin-phenazine-based polymers for gold recovery from authentic e-waste, which could achieve 99% uptake within 30 min and had a recovery capacity as high as 1.62 g/g towards $Au^{3+}$ with redox-active capability; and Ding et al.[16] synthesized a polyamine-functionalized porous organic polymer adsorbent (Pc−POSS−POP) by a hypercrosslinking pathway, with a peak gold adsorption capacity of 1026.87 mg g$^{-1}$ under light irradiation. Despite good performance in gold recovery from ore mines/e-waste, the tedious synthetic procedures of these materials and their high-cost precursors for metal recovery impede their further applications.

In line with their traditional descriptions as reactive or binding sites in chemical catalysis, N-heterocyclic carbenes and their derivatives have been applied as emerging material platforms, e.g., organic light-emitting diodes[17] and oxygen species-responsive reagents[18]. As a step forwards, "polycarbene", i.e., local assemblies of individual carbene units, have been introduced as versatile materials applied widely in photoluminescence, membrane separation, etc.[19–23]. To upgrade their functions, porous structures have been combined with carbenes to craft intriguing materials as functional porous organic cages and/or actuators in material applications[21,24]. Consequently, the synergistic interplay between pores and carbene leads to the emerging materials of porous carbene or polycarbene and is presumed to be a highly effective "nanotrap" for small molecules and ions. On the basis of previous[23], we incorporated this porous carbene concept into an alternative model system to determine if it could support gold metallurgy under ambient conditions.

In this study, we report a stable, easy-to-construct porous organic polycarbene (POPcarbene) network for high-performance gold extraction from waste electronic materials. By taking advantage of an ammonia-catalysed molecular crosslinking mechanism, poly(1,2,4-triazolium)s as polycarbene precursors were processed into a covalently locked porous polymer. The goal was to use the as-synthesized POPcarbene adsorbent to capture gold in an efficient and selective way, including gold in a complex multi-ionic solution containing $Au^{3+}$, $Pt^{2+}$, $Cu^{2+}$, $Mg^{2+}$, $Cr^{2+}$, $Zn^{2+}$, $Co^{2+}$, and $Ni^{2+}$. The ppb level of gold ion uptake in an aqueous solution was also considered to determine whether the adsorbent could enrich a trace amount of gold or not. We also revealed the mechanism of gold metallurgy by proving the formation of Au-carbene bonds via X-ray photoelectron spectroscopy (XPS) analysis and nuclear magnetic resonance (NMR) spectroscopy. Moreover, density functional theory (DFT) calculations were applied to increase our understanding of gold extraction selectivity and mechanisms. Finally, the production of this POPcarbene adsorbent was subjected to an LCA and cost analysis to determine its environmental impact and cost when producing such adsorbents at scale.

## Results
### Synthesis and characterization of Ptriaz-CN adsorbent (Ptriaz-CN-A).
The chemical synthesis of the 1,2,4-triazolium-based polymer poly[4-cyanomethyl-1-vinyl-1,2,4-triazolium bis(trifluoromethane sulfonyl)imide] (termed "Ptriaz-CN") is described in "Methods" and Fig. S1[23]. Its chemical structure and apparent molecular weight were characterized and confirmed by proton nuclear magnetic resonance ($^1$H NMR) spectroscopy and gel permeation chromatography (GPC, Figs. S2–S3). The apparent number-average molecular weight of Ptriaz-CN was 35,739 g mol$^{-1}$, and its polydispersity index was 2.13, as determined by GPC. Figure S5 shows the synthetic pathway towards the mesoporous Ptriaz-CN adsorbent (termed "Ptriaz-CN-A"). In a typical synthesis, the precursor solution was prepared by dissolving 100 mg Ptriaz-CN powder into 1 mL EtOH/DMF mixture solution (v/v = 17/83). The polymer mixture solution was heated at 80 °C for 4 h and then dripped into a vessel to cool at room temperature for 12 h, which led to thermally induced phase separation (TIPS) of the polymer in the form of viscous polymer solutions[25]. Next, the vessel was put into a 0.5 wt% $NH_3$ solution in EtOH, where the viscous polymer liquid gradually became a brown gel. These changes in colour and state were related to the cation-methylene-nitrile functionality sequence in the repeating unit of this poly(ionic liquid), in which the pendent nitrile (−CN) groups could undergo coupling reactions; this reaction was catalysed at room temperature by $NH_3$[26] and was followed by cyclization of −CN triple bonds to crosslink the entire polymer network, which, in combination with the TIPS process, generated pores. The formed polymer gels were subsequently soaked in EtOH to remove other embedded reagents and water molecules and then dried by supercritical $CO_2$ fluid to form a porous adsorbent. To investigate the surface properties of Ptriaz-CN-A, a water contact angle measurement was performed. Figure S6 shows that the surface layer initially had a water contact angle of ~60°, and after 6 min, a spontaneous infiltration process occurred, indicating that the adsorbent in contact with water became superhydrophilic and underwent a reversible infiltration process. Scanning electron microscopy (SEM) characterization also revealed that the structure of Ptriaz-CN-A barely changed during water infiltration (Fig. S7). Furthermore, the chemical process was also considered and evidenced by time-dependent $^1$H NMR. The results proved that the C5 proton was highly active, and reversible proton exchange took place between $H_2O$ and Ptriaz-CN-A, resulting in a time-dependent interaction between Ptriaz-CN-A and water (Fig. S8). The porosity was determined as it allows high-density packing of carbene units for metal binding, an essential factor governing the gold-capturing efficiency of the adsorbent. Hence, $N_2$ sorption (at 77 K) measurements based on the Brunauer–Emmett–Teller (BET) equation were performed to determine the pore characteristics of Ptriaz-CN-A. The specific BET surface area was calculated to be 332 m$^2$ g$^{-1}$ (Fig. 1c), which is fairly acceptable for porous materials prepared free of external templates. The nonlocal density functional theory method was employed to calculate the pore width[27,28]. The pore size of Ptriaz-CN-A was mainly distributed between 2 and 32 nm, indicating that the majority of pores were mesopores. SEM characterization revealed the distinctive morphology of mesoporous Ptriaz-CN-A (Fig. 1d). Particles 30−50 nm in size were clearly observed as a secondary structure motif, stemming from $NH_3$-triggered, intramolecularly crosslinked Ptriaz chains via trimerization of the nitrile groups in the cation-methylene-nitrile sequence of Ptriaz-CN. To further explore the chemical structure of the crosslinked network, we first took an unpolymerized precursor, the TriazoleTFSI monomer, and $NH_3$-treated TriazoleTFSI (termed TriazoleTFSI-$NH_3$) to provide sufficient spectroscopic evidence for the formation of networks. As shown in the Fourier transform infrared (FT-IR) spectra of the two substances (Fig. S9a), compared with untreated TriazoleTFSI, the appearance of new absorption bands located at 1672 cm$^{-1}$ (C=N) and 1225 cm$^{-1}$ (C−N) supported the cyclization reaction of nitrile groups into the s-triazine ring. In addition, the TriazoleTFSI-$NH_3$ monomer also gave rise to stretching vibration peaks of C=N (1610 cm$^{-1}$) for amidine and protonated amidine (1642 cm$^{-1}$)[29]. Furthermore, the $^{13}$C NMR spectra of typical TriazoleTFSI with or without

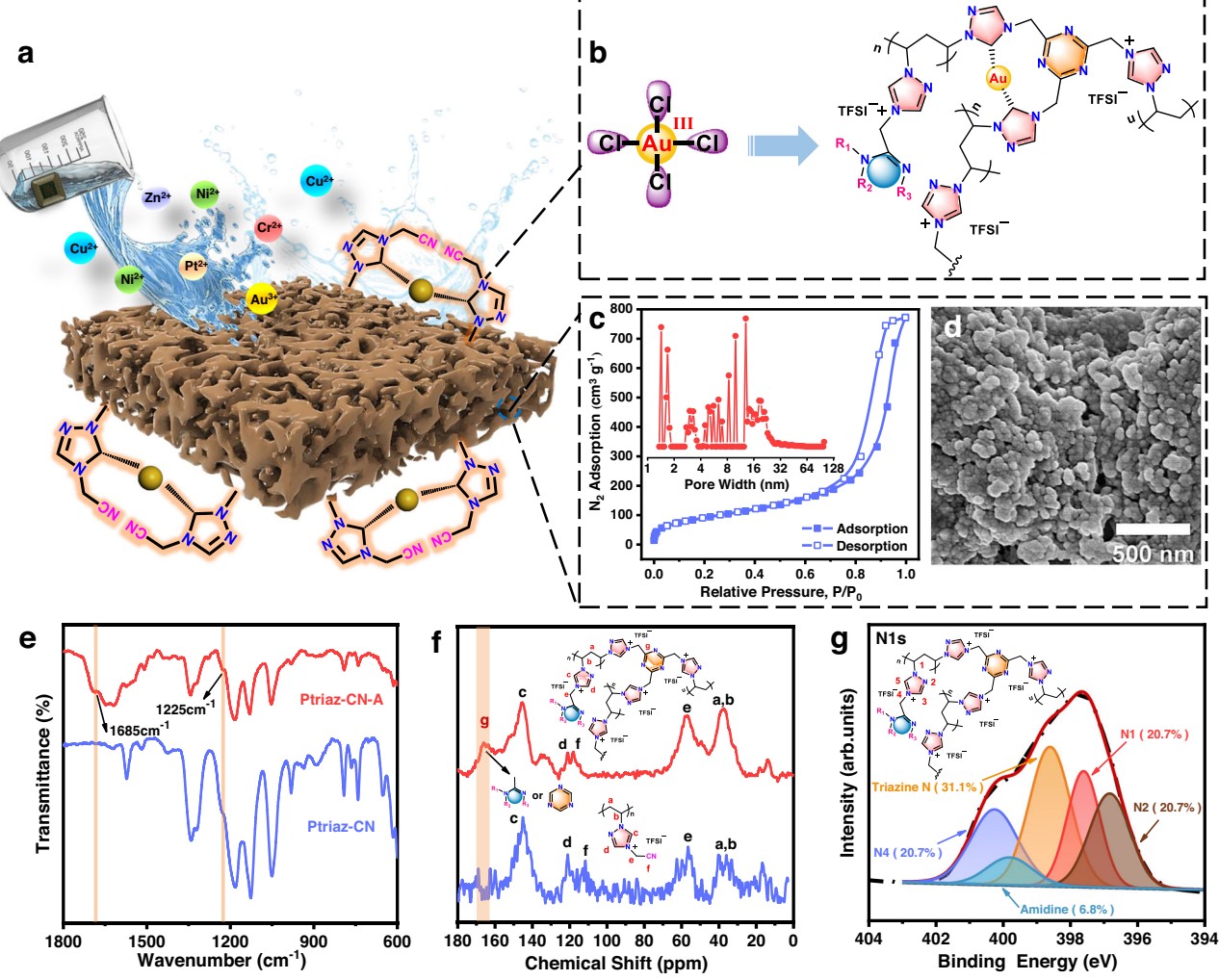

**Fig. 1 | Schematic illustration of the 1,2,4-triazolium-based POPcarbene adsorbent (Ptriaz-CN-A) for gold recovery and characterization of Ptriaz-CN and Ptriaz-CN-A. a** Scheme of gold recovery from electronic wastes by Ptriaz-CN-A via a wet-chemistry approach. **b** The mechanism of interaction between Au³⁺ and crosslinked Ptriaz-CN-A. **c** Nitrogen adsorption-desorption isotherms and pore size distribution plot of Ptriaz-CN-A. **d** SEM image of Ptriaz-CN-A. **e** FT-IR and **f** solid-state ¹³C NMR spectra of Ptriaz-CN (bottom line) and the as-synthesized Ptriaz-CN-A (top line). **g** N 1s XPS analysis of the as-synthesized Ptriaz-CN-A. The coloured shadings represent the internal integral area of different characteristic peaks. N4 (purple), Amidine (blue), Triazine (orange), N1 (red) and N2 (brown). Source data are provided as a Source Data file.

ammonia treatment were obtained. As shown in Fig. S9b, carbon signals at 163 ppm, 166–169 ppm and 170 ppm were observed for ammonia-treated TriazoleTFSI-NH₃ but were absent in pristine TriazoleTFSI, which belongs to protonated amidine, conventional amidine and *s*-triazine, respectively[29–31]. The above results prove that both *s*-triazine and amidine structures exist in ammonia-treated TriazoleTFSI. However, unlike the monomer TriazoleTFSI, in Ptriaz-CN (the polymerization product of TriazoleTFSI), the characteristic peaks of amidine in the FT-IR and solid-state ¹³C NMR spectra were not obvious and could not be directly distinguished from other signals. In contrast, the characteristic peaks of *s*-triazine were obviously seen and could be easily distinguished (Fig. 1ef)[26,32]. These results provided evidence that due to the "cation-methylene-nitrile" sequence, the polymer Ptriaz-CN became crosslinked via nitrile cyclization under ammonia treatment to form *s*-triazine structures that covalently crosslinked the polymer[26]. Moreover, according to the evidence we obtained from the tests performed on the monomer TriazoleTFSI, the amidine structures in the crosslinked polymers after ammonia treatment also could not be simply ruled out. As a consequence, we believe that both *s*-triazine and amidine structures coexist in Ptriaz-CN-A, although the *s*-triazine structure was dominant in the polymer product. From the N 1s XPS

characterizations of Ptriaz-CN-A (Fig. 1g), three distinct peaks at 400.1, 397.6 and 396.8 eV were observed and attributed to N atoms at position 4 (N4) and position 1 (N1) and bare N at position 2 (N2), respectively, in the 1,2,4-triazolium cation, with a 1:1:1 area ratio. Other peaks at 399.8 eV and 398.6 eV were attributed to N atoms in the amidine and *s*-triazine crosslink units (Triazine N), respectively[33,34]. On the basis of the XPS results, the crosslinking degrees of the nitrile groups to form *s*-triazine rings and amidine derivatives were calculated to be 31.1% and 6.8%, respectively. This indicates that one out of every three or four nitrile groups was uncoordinated in the *s*-triazine ring, which is fairly high, as each crosslinked nitrile group disabled adjacent nitrile units from ring formation due to steric hindrance.

### Adsorption performance towards metal adsorption and recovery

The adsorption efficiency of Ptriaz-CN-A towards precious metal ions was assessed based on the metal ion adsorption isotherms in various solutions of Au³⁺ or Pt²⁺ at different concentrations (100-5000 ppm). After stirring for 24 h until adsorption equilibrium was reached, the solutions were filtered off, and the filtrate was analysed via inductively coupled plasma–mass spectrometry (ICP–OES) to determine

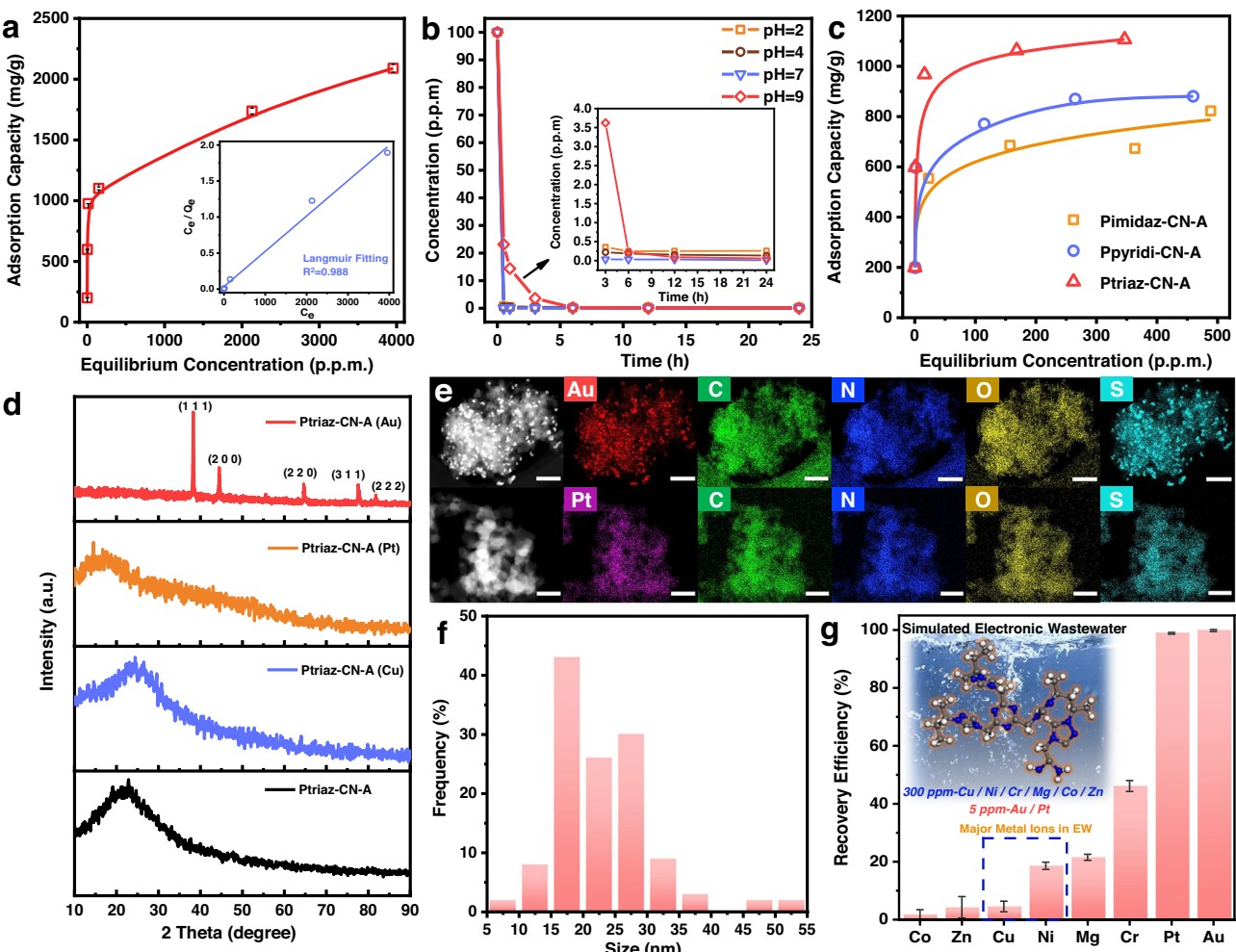

**Fig. 2 | Adsorption performance of Ptriaz-CN-A. a** Adsorption isotherm of Au³⁺ on Ptriaz-CN-A and related Langmuir isotherm fitting ($R^2 = 0.988$) **b** Time-dependent adsorption efficiencies of Au³⁺ at various pH values (pH = 2 (orange square), pH = 4 (brown circle), pH = 7 (purple triangle), pH = 9 (red rhombus)). **c** Au³⁺ adsorption isotherms of Ptriaz-CN-A (red triangle), Pimidaz-CN-A (orange square) and Ppyridi-CN-A (purple circle). **d** XRD patterns for Au-loaded, Pt-loaded, Cu-loaded, and metal-free Ptriaz-CN-A. **e** STEM images and elemental mapping for Au-loaded Ptriaz-CN-A (upper row, scale bar, 200 nm) and Pt-loaded Ptriaz-CN-A (lower row, scale bar, 100 nm). **f** The corresponding size distribution histogram of Au nanoparticles. **g** Selective adsorption of Au³⁺ on Ptriaz-CN-A in simulated electronic wastewater. Error bars represent standard deviation. $n = 3$ independent experiments. Source data are provided as a Source Data file.

the residual Au³⁺ and Pt²⁺ concentrations. Figure 2a and Fig. S10 show the adsorption isotherms of Au³⁺ and Pt²⁺ on Ptriaz-CN-A. The adsorption amounts increased with increasing Au³⁺/Pt²⁺ concentration, and the adsorption capacity greatly increased and reached a maximum at high concentrations. Ptriaz-CN-A seized significant amounts of gold ions, with an uptake capacity as high as 2.09 g/g, thus outperforming most reported porous organic polymers (Table S1). In contrast, in the same experiment, the adsorption amount of platinum in Ptriaz-CN-A was 1.62 g/g, nearly 78% of the gold uptake. Moreover, the Freundlich adsorption isotherm model and Langmuir adsorption isotherm model were employed to analyse the adsorption isotherm data. The results indicated that the adsorption isotherms were well fitted with the Langmuir model (Fig. 2a, Fig. S11 and Fig. S12), which yielded linear correlation coefficients as high as 0.988 for Au³⁺ and 0.88 for Pt²⁺.

Given the initial promising results for the uptake of gold, Ptriaz-CN-A was investigated in detail for kinetic efficiency under various pH values. With 5 mg of the adsorbent in 10 mL of Au³⁺ solution (100 ppm) under gradually increasing pH values from 2 to 9, the supernatant was collected at increasing time intervals. The filtrates were analysed by ICP–OES to detect the remaining Au³⁺ content in solution, and the concentration changes were measured over time. The adsorption

behaviour of Ptriaz-CN-A exhibits a weak correlation with the pH value of treated solutions since Ptriaz-CN-A is a cationic adsorbent, and its properties (electrostatic interactions with AuCl₄⁻) do not change in acidic or alkaline solutions. When the pH varied from acidic (pH = 2) to basic (pH = 9), the uptake activity of Ptriaz-CN-A declined, with a rapid adsorption rate at pH = 2, 4, and 7 (over 90% after 30 min) and a diminutive uptake (76.9%) at pH = 9 after 30 min (Fig. 2b). Nevertheless, the Au³⁺ concentration decreased to low to none after 12 h under any arbitrary pH value.

To obtain further insight into the adsorption capability of 1,2,4-triazolium-derived carbene species, their porous polymer counterparts with imidazolium and pyridinium groups (termed Pimidaz-CN and Ppyridi-CN, respectively) were applied as references for gold uptake. The chemical syntheses of Pimidaz-CN and Ppyridi-CN are described in the Methods. A fabrication process similar to that of Ptriaz-CN-A was conducted on the Pimidaz-CN adsorbent (termed Pimidaz-CN-A) and Ppyridi-CN adsorbent (termed Ppyridi-CN-A). The adsorption efficiencies of Ptriaz-CN-A, Pimidaz-CN-A and Ppyridi-CN-A were assessed based on the gold adsorption isotherms for various gold solutions of different concentrations at 100, 300, 500, 700, and 900 ppm. As shown in Fig. 2c, in the same concentration range, Ptriaz-CN-A exhibited the highest adsorption capacity (1.107 g/g), which is much

higher than that of Pimidaz-CN-A (0.8225 g/g) and Ppyridi-CN-A (0.8804 g/g).

The remarkable uptake and selectivity for $Au^{3+}$, even at elevated concentrations, prompted us to study Ptriaz-CN-A more comprehensively. The adsorbent samples before and after the $Au^{3+}$ capture tests were characterized by X-ray diffraction (XRD), transmission electron microscopy (TEM), XPS, and NMR. As shown in Fig. 2d, the XRD patterns of Au-loaded Ptriaz-CN-A displayed distinct peaks that fit well with metallic gold, indicative of the reductive property of the active N-heterocyclic carbene sites in the adsorption process; in comparison, Pt-loaded Ptriaz-CN-A, Cu-loaded Ptriaz-CN-A and Au-loaded Pimidaz-CN-A (Fig. S13) exhibited merely an amorphous halo. The in situ reduction of $Au^{3+}$ into metallic gold by Ptriaz-CN-A was further verified by gold nanoparticles 15–30 nm in size observed in the TEM images (Fig. 2e, f and Fig. S14), followed by their elemental mappings. In contrast, the adsorbed platinum and copper ions were particularly well dispersed in Ptriaz-CN-A, with no metal particles observed in the SEM or TEM images (Figs. S15–S16).

As previously mentioned, Ptriaz-CN-A was designed for gold recovery from an e-waste-derived aqueous solution that contained a variety of metal ions, some of which may compete with $Au^{3+}$ for adsorption sites due to entropic reasons. Thus, it was crucial for the designed adsorbent to adsorb specific metal ions, i.e., $Au^{3+}$. The selectivity of Ptriaz-CN-A for $Au^{3+}$ in the presence of interfering ions in a simulated e-waste solution was tested thereafter. To simulate realistic electronic wastewater more practically, we prepared e-waste solutions containing multiple interfering ions, including $Co^{2+}$, $Zn^{2+}$, $Cu^{2+}$, $Ni^{2+}$, $Mg^{2+}$, and $Cr^{2+}$, with concentrations of 300 ppm, 60 times higher than that of the target $Au^{3+}$ and $Pt^{2+}$. Typically, 5 mg of Ptriaz-CN-A was placed in a 10 mL mixed metal ion solution, and after 6 h, the remaining concentrations in the solution were measured by ICP–OES. Figure 2g shows the selective adsorption of $Au^{3+}$ on Ptriaz-CN-A in the presence of other highly concentrated metal ions in aqueous solution. As a result, Ptriaz-CN-A exhibited nearly full $Au^{3+}$ recovery (recovery efficiency (REE) of 99.8%), leaving a $Au^{3+}$ concentration down to 10 ppb. A similar adsorption preference was found for another precious metal ion, $Pt^{2+}$, with an REE of up to 98.9%. In the case of other metal ions, such as $Cr^{2+}$, although the REE of Ptriaz-CN-A for $Cr^{2+}$ was 46.1%,

the original concentration of $Cr^{2+}$ in electronic wastewater was much lower in the real case. Moreover, the REEs of Ptriaz-CN-A for $Mg^{2+}$ and $Co^{2+}$ were 21.5% and 1.75%, respectively. For major metal ions ($Cu^{2+}$ and $Ni^{2+}$) in the e-waste solution, Ptriaz-CN-A showed a weak adsorption effect on $Cu^{2+}$ and $Ni^{2+}$, with REEs of 4.53% and 18.6%, respectively, despite their 60 times higher concentrations compared with that of $Au^{3+}$. The results suggested a remarkable anti-interference ability of Ptriaz-CN-A for $Au^{3+}$ adsorption. Next, the affinity of Ptriaz-CN-A for $Au^{3+}$ was quantified by comparing the distribution coefficients ($K_d$, mL/g) of different ions[35]. As shown in Table S3, Ptriaz-CN-A had the largest distribution coefficient for $Au^{3+}$, and the $K_d$–based ion selectivity coefficient was ~6 for $Au^{3+}/Pt^{2+}$, $2.2 \times 10^3$ for $Au^{3+}/Ni^{2+}$, $1.1 \times 10^4$ for $Au^{3+}/Cu^{2+}$, $2.8 \times 10^4$ for $Au^{3+}/Co^{2+}$, $1.1 \times 10^4$ for $Au^{3+}/Zn^{2+}$, $0.6 \times 10^3$ for $Au^{3+}/Cr^{2+}$, and $1.8 \times 10^3$ for $Au^{3+}/Mg^{2+}$. These data prove the preferential uptake of Ptriaz-CN-A to noble metal ions ($Au^{3+}$ and $Pt^{2+}$) over non-precious metals in e-waste solutions. Finally, the adsorbent's ability to concentrate trace amounts of $Au^{3+}$ from its dilute stream over time was tested. For this experiment, 5 mg of Ptriaz-CN-A was added to 50 mL of 100 ppb $Au^{3+}$ solution under 300 rpm stirring. Within 24 h, 95.2% of $Au^{3+}$ was taken up, and >99.9% was taken up within 48 h. Generally, Ptriaz-CN-A exhibited a high sensitivity towards $Au^{3+}$ (aq.) even under trace concentrations.

The next step was to extract gold as a minor fraction from authentic electronic waste, where the disused CPU was chosen as a representative of the e-waste. The scrap CPU contains an array of pins mainly composed of Au, Pt, Cu and Ni, as can be observed in the energy dispersive X-ray spectroscopy (EDS) mapping for one single pin in the scrap CPU (Fig. 3a and Fig. S17). Then, the metal pins were removed from the CPU to be processed in the leaching solution that consisted of an aqueous mixture solution of N-bromosuccinimide (NBS) and pyridine (Py) at a neutral pH (Fig. 3b). Compared to traditional methods that normally use lethal alkali cyanide or extreme acidic solutions, Yang's leaching route employs mild and environmentally benign conditions (neutral pH and chemicals of low toxicity) that can efficiently oxidize $Au^0$ in electronic waste into $Au^{3+}$ in a high yield and selectivity[36]. The removed metal scraps were treated for 4 days at room temperature, and the resulting e-waste solution was filtered and then acidified by HCl to pH = 2 for better preservation of the dissolved metal ions.

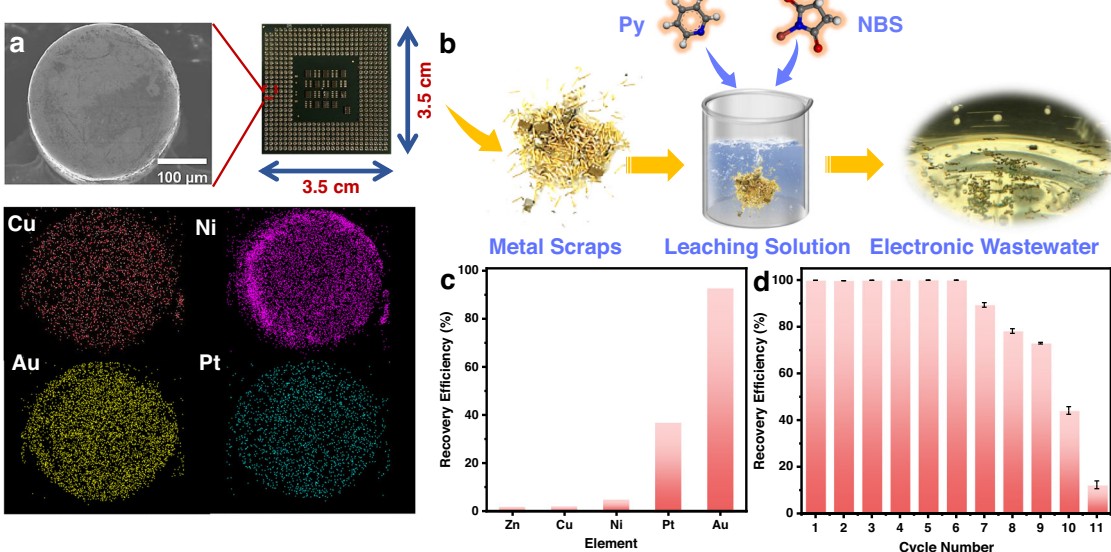

**Fig. 3 | Precious metal capture and regeneration test for authentic e-waste by Ptriaz-CN-A. a** SEM image and elemental mapping analysis of the metal pins (top view) on the CPU. **b** The e-waste-derived aqueous solution obtained by soaking metal scraps removed from CPU in a leaching solution. **c** Gold recovery efficiency from actual electronic waste solution using Ptriaz-CN-A. **d** Regeneration and reusability plots of Ptriaz-CN-A. The error bars represent standard deviation. $n = 3$ independent experiments. Source data are provided as a Source Data file.

The final leaching solution contained a number of elements, including $Au^{3+}$, $Pt^{2+}$, $Cu^{2+}$, $Ni^{2+}$ and $Zn^{2+}$ (see Table S4 for the concentration of metals). In this leaching solution, $Cu^{2+}$ was the major metal component (357.56 ppm), and $Au^{3+}$ was the minority component (2.235 ppm). Nevertheless, Ptriaz-CN-A displayed a satisfactory selectivity for $Au^{3+}$ (REE ~ 92.75%) and $Pt^{2+}$ (REE ~ 36.97%) in the presence of abundant $Cu^{2+}$ and other metal ions. In contrast, the $Cu^{2+}$ concentration in e-waste water was reduced by only ~8 ppm to 349.22 ppm, giving an ultralow REE of 2.33% (Fig. 3c). To study the properties of recovered gold nanoparticles after extraction from the e-waste solution, Ptriaz-CN-A (Au) was calcined in air at 900 °C, and the obtained powder was subsequently immersed in concentrated hydrochloric acid (*aq.*). Despite high concentrations of competing ions in e-waste solution (e.g., $Cu^{2+}$, $Ni^{2+}$), these competitive ions were easily dissolved in concentrated hydrochloric acid after soaking. After that, the treated powders were subsequently washed with deionized water and dried in an oven. Obviously, the XRD patterns of the resulting material displayed distinct peaks that fit well with metallic gold (Fig. S18). Furthermore, more characterization by SEM, TEM, STEM and elemental mapping of the recovered gold particles were also performed (Figs. S19–S21).

Efficient desorption of gold from spent Ptriaz-CN-A is essential to its reusability. An acidic thiourea solution was selected as the desorption agent because it can restore the effective adsorption sites largely if not entirely in the oxidized adsorbents[4]. The desorption test was carried out by applying the spent adsorbent, i.e., the Au-loaded Ptriaz-CN-A (termed Ptriaz-CN-A (Au)), which was saturated by $Au^{3+}$ in a 100 ppm $Au^{3+}$ solution. The desorption of Au utilizing elution solution contained a 1:1 (v/v) ratio of thiourea (1.0 mol/L) and HCl (1.0 mol/L) under stirring at 60 °C for 8 h. Figure 3d showed the reusability results of Ptriaz-CN-A after 11 cycles of $Au^{3+}$ recovery. The results demonstrate that the recovery efficiencies for Ptriaz-CN-A were all above 95% after six cycles, indicating its excellent reusability. The REE decreased continuously in the subsequent cycles owing to the loss of the free spaces in adsorbents due to the continuous enrichment of the reduced Au nanoparticles in the networks of Ptriaz-CN-A. Furthermore, the regenerated Ptriaz-CN-A still exhibited more than 60% elution efficiency (EEE) for $Au^{3+}$ after 11 adsorption-desorption cycles (Fig. S22).

## Adsorption mechanism

The XPS analysis of pristine and Au-loaded Ptriaz-CN-A is shown in Fig. 4a. Signals of C 1$s$, O 1$s$, N 1$s$, F 1$s$ and Au 4$f$ peaks can be clearly observed from the wide-range XPS spectra of Ptriaz-CN-A -Au. The fitted XPS spectra in Figs. 4b–e exhibit two oxidation states of Au, i.e., Au 4$f_{7/2}$ and Au 4$f_{5/2}$, after adsorption of $Au^{3+}$ in aqueous solutions at different concentrations by Ptriaz-CN-A. As shown in Fig. 4b and c, after adsorbing $Au^{3+}$ at lower concentrations of 10 ppm and 100 ppm, the peak assigned to Au 4$f$ was divided into two peaks corresponding

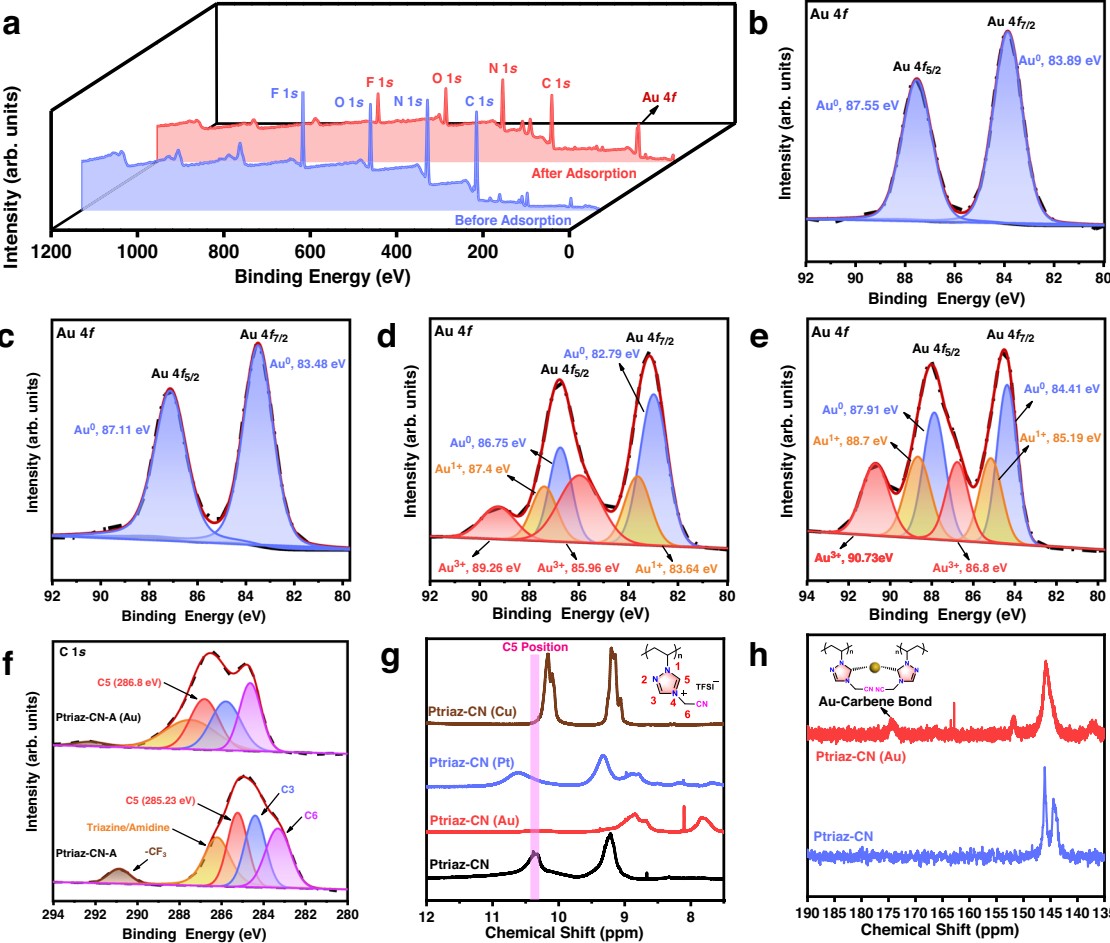

**Fig. 4 | Analysis of Ptriaz-CN-A binding interactions with metal ions. a** XPS profile of Ptriaz-CN-A before and after $Au^{3+}$ adsorption. **b**–**e** High-resolution Au 4$f$ XPS spectra after adsorption tests in 10, 100, 300 and 500 ppm $Au^{3+}$. The coloured shadings represent the internal integral area of different characteristic peaks. $Au^0$ (purple), $Au^{1+}$ (orange) and $Au^{3+}$ (red). **f** High-resolution C 1$s$ XPS spectra before and after $Au^{3+}$ adsorption. The coloured shadings represent the internal integral area of different characteristic peaks. C3 (purple), C6 (pink), $s$-triazine/amidine C (orange), C5 (red) and -CF$_3$ (brown). **g** $^1$H-NMR spectra of Ptriaz-CN and Ptriaz-CN (metal ion) complex. **h** $^{13}$C-NMR spectra of Ptriaz-CN and Ptriaz-CN (Au). Source data are provided as a Source Data file.

to Au[0]: Au $4f_{5/2}$ at binding energies of 87.55/87.11 eV and Au $4f_{7/2}$ at 83.89/83.48 eV[37,38]. However, at higher Au[3+] concentrations of 300 ppm and 500 ppm, the XPS spectra of Ptriaz-CN-A-Au showed three similar oxidation states of gold. At 300 ppm, these states were Au[0] at 86.75 ($4f_{5/2}$) eV and 82.79 ($4f_{7/2}$) eV, Au[1+] at 87.4 ($4f_{5/2}$) eV and 83.64 ($4f_{7/2}$) eV, and Au[3+] at 89.26 ($4f_{5/2}$) eV and 85.96 ($4f_{7/2}$) eV (Fig. 4d)[11], with abundances of 29.3%, 23.7%, and 47.0%, respectively. Likewise, at 500 ppm, the three oxidation states were Au[0] at 87.91 ($4f_{5/2}$) eV and 84.41 ($4f_{7/2}$) eV, Au[1+] at 88.7 ($4f_{5/2}$) eV and 85.19 ($4f_{7/2}$) eV, and Au[3+] at 90.73 ($4f_{5/2}$) eV and 86.8 ($4f_{7/2}$) eV (Fig. 4e)[39], with abundances of 27.73%, 28.73%, and 43.54%, respectively. In contrast, the Ptriaz-CN-A-Pt and Ptriaz-CN-A-Cu samples barely showed ionic state signals (Figs. S24–S25), in good agreement with their XRD analyses. These results confirm that Ptriaz-CN-A participated in a chemical redox reaction to reduce Au[3+] to Au[1+] and Au[0] during the adsorption process, and similar findings were previously reported for N-heterocyclic carbene units by other groups[40–43] Hence, to better understand the interaction between Au[3+] and Ptriaz-CN-A, the XPS C 1s spectra of Ptriaz-CN-A and Ptriaz-CN-A-Au were compared to study the actual active adsorption site of Ptriaz-CN-A for gold. As shown in Fig. 4f, the line shapes of the C 1s spectra showed obvious differences before and after the loading of gold ions onto Ptriaz-CN-A. The C 1s signals at 290.86, 286.2, 285.23, 284.39 and 283.31 eV were assigned to -CF₃, s-triazine/amidine C, C5, C3 and C6, respectively[26,44]. After the adsorption of Au[3+], the binding energy for C5 shifted significantly from 285.23 to 286.8 eV, while the binding energy for -CF₃ and s-triazine/amidine C shifted slightly from 290.86 eV and 286.2 eV to 292.2 eV and 287.47 eV, respectively, indicating the distinct attribution of C5-metal (metal-carbene) interactions to the adsorption behaviour. In comparison, the characteristic peaks for C5 in the 1,2,4-triazolium unit changed only slightly after adsorbing Pt[2+] (from 285.23 to 285.30 eV) and Cu[2+] (from 285.23 to 285.19 eV), indicating negligible interactions between the C5 site in Ptriaz-CN-A and platinum and copper.

To specify the binding site of Ptriaz-CN-A, NMR measurements were employed. Figure 4g displays the ¹H-NMR spectra of Ptriaz-CN and the Ptriaz-CN/metal ion complex in a H₂O/DMSO-d₆ mixture (v/v = 1:5) solvent. The proton signals of C3 and C5 in the 1,2,4-triazo-lium ring were clearly observed at 9.2 and 10.4 ppm, respectively. Upon the addition of Au[3+], the C5-proton signal in the 1,2,4-triazolium ring disappeared, indicative of its reductive and coordinative interplay with Au[3+]. Moreover, in the ¹³C-NMR spectra of Fig. 4h, the representative resonance of the C-Au bond was detected at 174 ppm, supporting that the C5/carbene site has a strong coordination effect with gold[45]. For the ¹H-NMR spectra of the Ptriaz-CN/(Pt[2+] and Cu[2+]) complex solutions, the C5-proton signals in the 1,2,4-triazolium ring shifted slightly from 10.4 ppm to 10.6 ppm and 10.1 ppm, indicating H···Pt[2+] and H···Cu[2+] interactions, respectively, instead of reductive coordination. Importantly, metal-carbene peaks were not observed in the ¹³C-NMR spectra for both Ptriaz-CN/(Pt[2+] and Cu[2+]) complex solutions (Fig. S26). These results prove that metal ions other than Au[3+] interacted with the C5-proton as a dynamic Lewis acid-base pair so that the C5-proton could not be eliminated to generate the carbene site for coordination with metal ions to form metal-carbene bonds.

In consideration of the aforementioned experimental analyses, the adsorption-reduction process and metal-carbene bond formation for capturing gold were investigated. In the 1,2,4-triazolium cation ring, the C5 proton was highly active and underwent easier deprotonation than other N-heterocyclic cations, e.g., imidazolium[46]. During the process of deprotonation and coordination with metal ions, reductive elimination between 1,2,4-triazolium and Au[3+] in the ionic form of AuCl₄⁻ occurred with stepwise reduction of Au[3+] to Au[1+] and Au[0] by releasing HCl molecules, wherein the N-heterocyclic moiety acted as an electron-donating group that increased the electron density of Au[0] and further stabilized the subsequently generated gold nanoparticles[47,48]. Our data support a viable adsorption-reduction

pathway as follows: Ptriaz-CN-A adsorbs free Au[3+] in solution, and the N-heterocyclic carbene sites coordinate with gold ions next to form C-Au bonds and generate reductive immobilization of Au[3+] into Au[0]. After the chemical reduction realized by deprotonation at the C5 position in the 1,2,4-triazolium ring, nearby Au[0] species aggregated spontaneously to form larger Au nanoparticles in the hierarchically porous Ptriaz-CN-A.

## Computational simulations

To reveal the growth mechanism of gold nanoparticles, we proposed a mechanism of Au[3+] adsorption-reduction on Ptriaz-CN-A (Fig. 5a) and performed DFT calculations to simulate the binding interactions between Ptriaz-CN-A and AuCl₄⁻/PtCl₄²⁻. The corresponding adsorption and binding energies were denoted as $E_{ads}$ and $E_{bind}$, respectively (Supplementary Section 3. Theoretical calculations). Taking the first $E_{ads}$ as an example, it was calculated as the energy released by absorbing the free metal ions in aqueous solution (Fig. 5b). The second binding energy, $E_{bind}$, was calculated by removing two Cl⁻ ions of AuCl₄⁻ and/or PtCl₄²⁻ to produce two HCl molecules, with the assistance of protons in the C5 position of 1,2,4-triazolium rings as the reducing agent. The remaining Cl⁻ was removed to generate one HCl molecule with additionally supplied H atoms in aqueous solution (H₃O⁺). Furthermore, some critical values (chemical hardness, chemical potential, etc.), molecular electrostatic potential diagrams and HOMO/LUMO representations in the adsorption process were also calculated by Multiwfn software[49,50] (Tables S5, S6, Figs. S32–S35).

The $E_{ads}$ of Ptriaz-CN-A with PtCl₄²⁻ ions was calculated to be −17.42 eV based on DFT simulation. In the poly(1,2,4-triazolium)-Pt model, although the binding energy in the formation of the Pt-carbene bond is −7.08 eV, which indicates its stable coordination state, the energy difference (ΔE) in the reaction process is positive (0.54 eV) during the removal of two Cl⁻ and the formation of the metal-carbene bond. This result indicates the thermodynamic infeasibility of Pt-carbene bond formation between 1,2,4-triazolium and platinum ions at room temperature in this specific case, despite the negligible energy difference in step 2. A positive energy difference was also observed for CuCl₂ adsorption in the reaction process of Cu-carbene bond formation (Fig. S36). In comparison, a stepwise exothermic process was observed from the calculated Au-carbene binding energies. In detail, the 1,2,4-triazolium units first adsorbed neighbouring AuCl₄⁻ from the aqueous solution, and the DFT calculated energy difference (ΔE1) was −8.16 eV (step 1). Then, two HCl molecules were removed from acid-base neutralization between the protons in the C5 position of the 1,2,4-triazolium units with Cl⁻ in AuCl₄⁻, followed by Au-carbene bond formation between the remaining AuCl₂⁻ and the C5/carbene sites (step 2). This energy difference (ΔE2) was calculated to be −10.61 eV, leading to a stable configuration of 1,2,4-triazolium unit-AuCl₂⁻ ($E_{bind}$ = −7.28 eV). Then, the remaining Cl⁻ in AuCl₂⁻ was further removed by generating HCl molecules, resulting in an energy difference (ΔE3) of −12.25 eV. Afterwards, AuCl₂⁻ was chemically reduced to Au, which coordinated strongly with 1,2,4-triazolium units with $E_{bind}$ = −4.62 eV (step 3). The computational simulations were clearly consistent with the experimental data, justifying the stronger adsorption capability of gold than platinum. Moreover, to understand the formation mechanism of gold nanoparticles, the binding process between Au[0] atoms was also simulated. As shown in Fig. 5b and c, the energy difference (ΔE4) was calculated to be −2.18 eV, indicating the thermodynamically feasible aggregation of Au[0] to form larger nanoparticles at room temperature (step 4). Moreover, regardless of whether the solvent effect is applied in DFT calculations, the trend of the reaction process was the same (Tables S7–S10). These results elaborate the thermodynamic feasibility of Au-carbene bond formation between 1,2,4-triazolium and gold ions at room temperature and confirm the spontaneous adsorption-reduction process of Au on Ptriaz-CN-A.

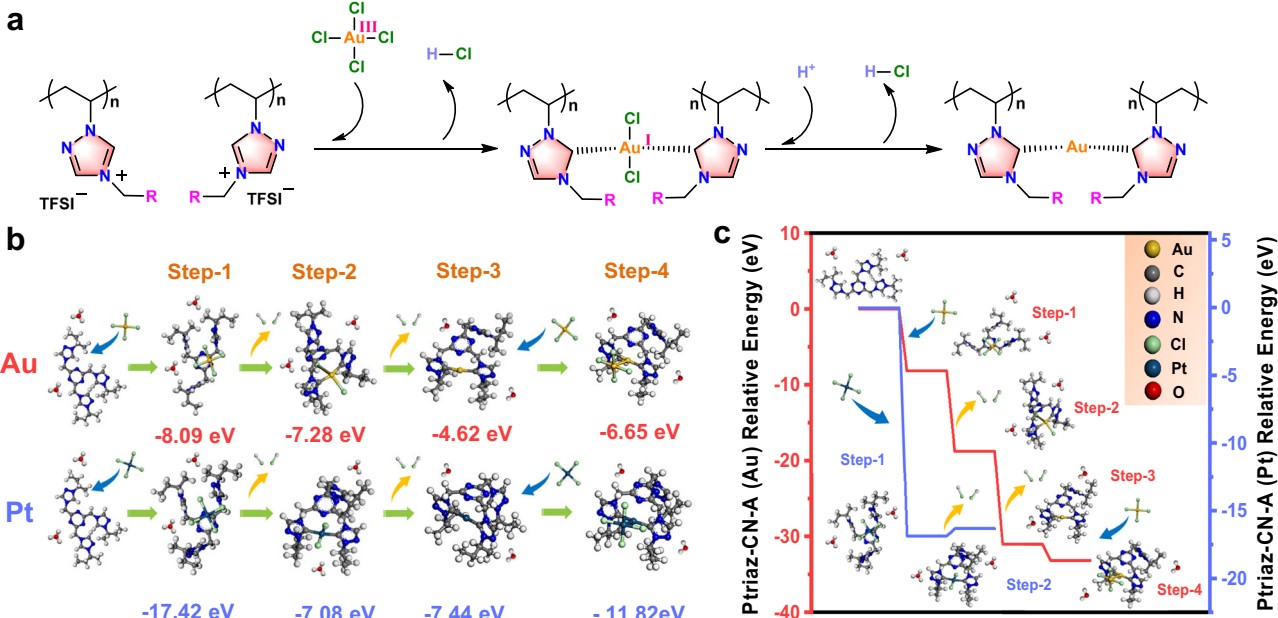

**Fig. 5 | DFT calculation analysis. a** Proposed mechanism of Au³⁺ adsorption-reduction on Ptriaz-CN-A. **b** Adsorption energy ($E_{ads}$) in step 1 and binding energy ($E_{bind}$) in sequential step 2 to step 4 of $AuCl_4^-$ and $PtCl_4^{2-}$ with a model unit of Ptriaz-CN-A. **c** DFT calculated energy differences in the adsorption process of $AuCl_4^-$ and $PtCl_4^{2-}$ on a model unit of Ptriaz-CN-A. Source data are provided as a Source Data file.

## Life cycle assessment (LCA) and cost analysis

LCA is a systematic tool for determining the environmental impact of a product or a process across its entire life cycle or a portion of its life cycle[51,52]. Therefore, the environmental impacts of Ptriaz-CN-A production prepared at different scales in this research were analysed using the "cradle-to-gate" LCA approach (the detailed data and analysis are summarized in Supplementary Section 4. Streamlined LCA). Figure S37 shows the system boundary for this LCA study, and all data were at the lab scale. As shown in Fig. 6a, it was clear that Ptriaz-CN-A prepared at Scale-2 had much lower environmental impacts across all categories. This is because equipment can be operated under full load conditions and resources can be maximized. As a consequence, there was much room to reduce the environmental impact of Ptriaz-CN-A production when it was produced on an industrial scale.

Cost analysis of raw materials and electricity in the Ptriaz-CN-A preparation process in Scale-1 and Scale-2 are presented in Fig. 6b (the detailed data and analysis are summarized in Supplementary Section 5. Cost analysis). The production costs for the synthesis of 1 g Ptriaz-CN-A in Scale-1 and Scale-2 were ~117.0 CNY and 107.3 CNY, respectively, more than 95% of which was spent on raw materials. Although this production cost seems high, it should be noted that the cost here was based on our laboratory data. It was clear that the cost preparation is reduced when we scale up production under laboratory conditions (Scale-2). Therefore, the preparation cost of this adsorbent can be greatly reduced in industrial production. Furthermore, the value of the gold captured by 1 g of Ptriaz-CN-A was ~795.2 CNY (gold price: ~380.5 CNY/g), and the profit margin will continue to increase by regenerating Ptriaz-CN-A. In general, Ptriaz-CN-A contributes a green and sustainable method for gold extraction from e-waste solutions.

## Discussion

In summary, we report a 1,2,4-triazolium-based, environmentally friendly and easy-to-prepare POPcarbene adsorbent with efficient and selective gold stripping ability from e-waste. This material exhibited an ultrahigh gold uptake of 2.09 g/g, with a bundled ability to chemically reduce Au³⁺ to Au⁰. In addition, this POPcarbene adsorbent exhibited a high selectivity for Au³⁺ in authentic waste electronic solutions in the presence of Au³⁺, Pt²⁺, Cu²⁺, Ni²⁺, and Zn²⁺ and could extract Au³⁺ quantitatively (>95%) at the 100 ppb level from its aqueous solution. The distinguished performance of such POPcarbene adsorbent was ascribed to their strong metal-carbene binding affinity, as evidenced by XPS and NMR analysis. The adsorption processes of metal ions onto this porous material were calculated by DFT methods combined with experimental equilibrium data, which further confirmed the adsorption and spontaneous reduction mechanisms of Au³⁺ on porous polycarbenes. In addition, LCA results and cost analysis clearly indicated that this POPcarbene adsorbent contributed to a green and sustainable method for gold extraction from e-waste solutions. These results suggest that this POPcarbene adsorbent is a highly selective and efficient adsorbent for gold recovery processes and can be used as a powerful material platform. It is expected that this finding will strengthen efforts to expand carbene chemistry further to materials science and promote the development of porous carbene or porous polycarbene materials.

## Methods

### Materials

Hydrogen tetrachloroaurate trihydrate ($HAuCl_4 \cdot 3H_2O$, 99%) was purchased from the Shanghai Bide Pharmatech Ltd. 1-vinyl-1,2,4-triazole (98%), Potassium tetrachloroplatinate ($K_2PtCl_4$) was purchased from Sigma-Aldrich. Dimethylformamide (DMF, 99.5%), Dimethyl sulfoxide (DMSO, 99.5%), alcohol (EtOH, 99.7%) and methanol (MeOH, 99.9%), diethyl ether (99.7%), tetrahydrofuran (THF, 99%) were purchased from Sinopharm Chemical Reagent Co., Ltd. Liquid ammonia ($NH_3$, 7.0 M solution in MeOH), bis (trifluoromethylsulfonyl) imide lithium (LiTFSI, 99%), bromoacetonitrile (BrCN, 97%), 2,2'-Azobis(2-methyl-propionitrile) (AIBN, 98%), copric chloride dihydrate ($CuCl_2 \cdot 2H_2O$, AR), 2,6-di-tert-butyl-4-methylphenol (BHT, 98%) were purchased from Macklin Co., Ltd. All other chemicals were analytic grade and used without further purification.

### Synthesis of 4-cyanomethyl-1-vinyl-1,2,4-triazolium bromide

1-vinyl-1,2,4-triazole (5.5 g, 0.05783 mol), bromoacetonitrile (10.41 g, 0.086745 mol) and 2,6-di-tert-butyl-4-methylphenol (0.05 g, 0.227 mol)

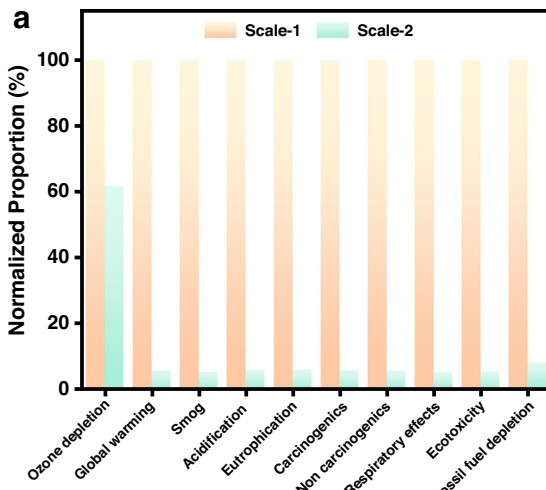

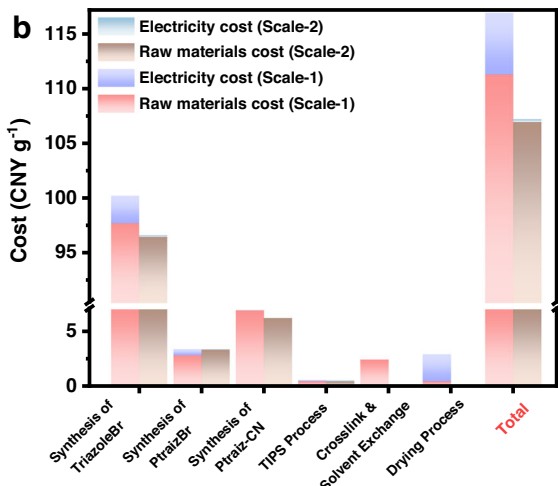

**Fig. 6 | LCA and cost analysis. a** Relative environmental impacts of 1 g Ptriaz-CN-A production at different preparation scales. The higher environmental impact in each category is normalized. **b** Production cost distribution of raw materials, electricity in each step for the synthesis of 1 g Ptriaz-CN-A at different preparation

scales (Scale-1: small dose feeding, data based on our current research; Scale-2: big dose feeding, data based on the maximum production scale of the laboratory). Source data are provided as a Source Data file.

were added into the flask. After the mixture was stirred for 72 h at 55 °C, the precipitate was filtered off and washed with diethyl ether and finally dried under vacuum at room temperature. (Yield: 95%, 11.80 g).

### Synthesis of poly(4-cyanomethyl-1-vinyl-1, 2, 4-triazolium bromide)

4-cyanomethyl-1-vinyl-1, 2, 4-triazolium bromide monomer (11.80 g, 0.05488 mol), AIBN (0.135 g, 0.8232 mmol), and 100 mL of DMSO were loaded into a 250 mL flask. The mixture was deoxygenated three times by a freeze-pump-thaw procedure and finally charged with nitrogen. The reaction mixture was then placed in an oil bath at 70 °C for 16 h. When cooling down to room temperature, the reaction mixture was dropwise added to an excess of tetrahydrofuran (THF). The precipitate was filtered off, washed with excess of ethanol and dried at 60 °C under vacuum. (Yield: 81%, 9.558 g).

### Synthesis of poly(4-cyanomethyl-1-vinyl-1,2,4-triazolium bis (trifluoromethanesulfonyl)imide) (termed Ptriaz-CN)

Anion exchange was performed by dropwise addition of solution 2) into solution 1): (1) 9.5 g of poly (4-cyanomethyl-1-vinyl-1,2,4-triazolium bromide) was dissolved in 490 mL deionized (DI) water, and (2) 1.2 eq. of bis (trifluoromethylsulfonyl) imide lithium salt solution. After addition, white precipitate appeared, the reaction mixture was allowed to stir for 6 h, and the precipitate was filtered, washed several times with DI water and dried in vacuum at 60 °C. (Yield: 98%, 17.89 g).

### Synthesis of poly(3-cyanomethyl-1-vinyl-imidazolium bis (tri-fluoromethanesulfonyl)imide) (termed Pimidaz-CN)

Synthesis of poly(3-cyanomethyl-1-vinyl-imidazolium bis (tri-fluoromethanesulfonyl)imide) (termed Pimidaz-CN) was performed according to the published method[26]. Typically, for synthesis of 4-cyanomethy-1-vinylimidazole bromide (termed as ImidazBr), 1-vinylimidazole (5.5 g, 0.0585 mol), bromoacetonitrile (8.424 g, 0.0702 mol) and 2,6-di-tert-butyl-4-methylphenol (0.05 g, 0.227 mol) were added into the flask. After the mixture was stirred for 24 h at 60 °C, the precipitate was filtered off and washed with THF for three times and finally dried under vacuum at 60 °C for 12 h. For synthesis of poly(4-cyanomethyl-1-vinyl-imidazole bromide) (termed as Pimi-dazBr), ImidazBr (10 g) and AIBN (80 mg) were dissolved in 120 mL DMSO. The mixture was deoxygenated three times by a freeze-pump-thaw procedure and finally charged with nitrogen. The reaction

mixture was then placed in an oil bath at 60 °C for 12 h. Afterwards the solution was dropped into excessive THF, and the precipitates (Pimi-dazBr) were collected, washed with THF three times and vacuum dried at for 60 °C overnight. For synthesis of poly(3-cyanomethyl-1-vinyl-imidazolium bis (trifluoromethanesulfonyl)imide) (termed Pimidaz-CN), anion exchange was performed by dropwise addition of solution 2) into solution 1), (1) 6 g of PimidazBr was dissolved in 300 mL deio-nized (DI) water, and (2) 1.2 eq. of bis (trifluoromethylsulfonyl) imide lithium salt solution. After addition, white precipitate appeared, the reaction mixture was allowed to stir for 6 h, and the precipitate was filtered, washed several times with DI water and dried in vacuum at 60 °C.

### Synthesis of poly(1-cyanomethyl-4-vinyl-pyridinium bis (trifluoromethanesulfonyl)imide) (termed Ppyridi-CN)

Synthesis of poly(1-cyanomethyl-4-vinyl-pyridinium bis (tri-fluoromethanesulfonyl)imide) (termed Ppyridi-CN) was performed according to the published method[53]. Typically, poly(4-vinyl pyridine) (5 g, 0.04755 mol) was dissolved in 60 mL of DMSO/ ethylene glycol (40 mL:20 mL). After purging the solution for 30 min with N₂, it was heated to 90 °C and bromoacetonitrile (17.13 g, 0.1423 mol) was added dropwise. The mixture was stirred at 90 °C overnight. Afterwards, the solution was dropped into THF, and the precipitates were collected and dialysed in water subsequently. After drying, 3 g (0.0133 mmol) of the as-obtained polymer were collected and re-dissolved in 250 mL of water for the anion exchange, 1.2 eq. of bis (trifluoromethylsulfonyl) imide lithium salt solution was dissolved in 10 mL of water and added dropwise to the stirred polymer solution. After addition, white pre-cipitate appeared, the reaction mixture was allowed to stir for 8 h, and the precipitate was filtered, washed several times with DI water and dried in vacuum at 80 °C.

## Data availability

The authors declare that all the data supporting the findings of this study are available within the article and Supplementary Information. Source data are provided with this paper.

## Code availability

Gaussian 09 (Revision D.01) for the DFT calculations is available at https://gaussian.com/glossary/g09/. Multiwfn (Version 3.8) for the DFT calculations is available at http://sobereva.com/multiwfn.

Simapro (Version 9.0) for the LCA analysis is available at https://simapro.com/

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

## Acknowledgements

We gratefully acknowledge the supports from the National Key Research and Development Program of China (2022YFB3807100, 2022YFB3807102, 2022YFB3807103), the National Natural Science Foundation of China (22102021, 52073046, 51873036 and 51673039), the Fundamental Research Funds for the Central Universities (2232022A-03), the Shanghai Shuguang Program (19SG28), the Program of Shanghai Academic Research Leader (21XD1420200), the Chang Jiang Scholar Program (Q2019152), the Shanghai Pujiang Program (20PJ1400300), the Natural Science Foundation of Shanghai (21ZR1402700 and 19ZR1470900) for the financial supports. The authors furthermore acknowledge the support from the International Joint Laboratory for Advanced Fiber and Low-Dimension Materials (18520750400).

## Author contributions

W.-Y.Z. and Y.-Z.L. conceived the idea and designed the experiments. X.-H.L. performed the experiments. M.-Q.Y., H.-Y.Z. and Z.-H.C. contributed to materials characterizations. Y.-L.W., L.-L.Z. and J.W. contributed to DFT calculations. X.-H.L. and R.L. contributed to LCA analysis. C.Q. and J.-Y.Y. joined the discussion of the data and gave helpful suggestions. All authors participated in drafting the paper, and gave approval to the final version of the manuscript. X.-H.L. and W.-Y.Z. wrote the manuscript.

## Competing interests

The authors declare no competing interests.
