## [Peer Review File · Nature Communications]

Porous organic polycarbene nanotrap for efficient and selective gold stripping from electronic wasteEditorial Note: Parts of this Peer Review File have been redacted as indicated to remove third-party material where no permission to publish could be obtained.

REVIEWER COMMENTS

Reviewer #1 (Remarks to the Author):

Li et al. reports gold capture by a network polymer made from a previously reported poly ionic liquid (Ptriaz, ref 23). The new polymer is made by treating Ptriaz with ammonia to make triazine crosslinkers. The solids were used in treating metal mixtures for selective metal uptake. The gold uptake reaches significant capacities of over 2 grams per gram polymer. Although the capacity and selectivity of Ptriaz towards gold is notable, there are major issues with the manuscript that need to be squarely addressed:

- How is it possible to make a material hydrophilic after 11 min of water contact? Is it because there is some hydrolysis? Is it reversible? There should be some chemical explanation. (Ln 121)
- The spectroscopic evidence for triazine formation is not adequate. It could simply be amidines that are forming, since ammonia reacting with nitrile normally produces amidine. Judging from the chelating activity of the adsorbent, it makes more sense that the structures are predominantly amidine networks. Low solubility is expected since amidines will have significant h-bonding. Also, the authors previous work (ref 23) employs ammonia to make membranes without any claim for triazine formation. This is confusing.
- If there's a reductive removal for gold, why is Pt adsorbed as much as 78% of gold? Pt uptake is much more than expected. What mechanism is at play for Pt?
- Cost issue is mentioned for introducing new materials but the synthesis is not short and uses costly reagents. It is not clear whether authors are making any gains there. There needs to be a LCA and process calculations to claim lowest cost.
- 100 ppb is not an "ultra-trace concentration".
- The porosity (BET area of 177 m²/g) is reported without any isotherms or calculations. Figure 1c is not showing any N₂ isotherms. BET range and calculations must be disclosed.

Formatting:

- Typos: has been comined, traizolium, non-prescious metals,
- Ptriaz-CN vs Ptriaz-CN. Please make sure you're consistent with codes.

Reviewer #2 (Remarks to the Author):

The comments to Authors can be found on the manuscript.

Reviewer #3 (Remarks to the Author):

This manuscript reports the design and preparation of polyionic liquid-derived porous organic polycarbene (POPCarbene) adsorbent and investigated its gold recovery performance from e-waste.

1. Although some results have been achieved, authors only do much routine work to follow the findings of previous reports, such as POP-pNH₂-py (Angew. Chem. Int. Ed. 2020, 59, 1-6) and Triazine-based POPs (Chemical Engineering Journal 2022, 430, 132618).
2. The morphology modulation of Ptriaz-CN-A is not fine enough and the resulting specific surface area is low.
3. As for the gold extraction from CPU, the mechanism of elution regeneration is needed for further investigation. Is there some variation in the microstructure of recycled materials? In addition, the number of cycling recovery is too low and it is recommended to increase to more than 10.
4. Generally speaking, an adsorbent with a K_d value × 10⁴ mL g⁻¹ is regarded to possess excellent adsorption performance. From Figure 3, K_d(Au) is only 998. Evidently, the proposed POPCarbene adsorbent does not possess superior selectivity, unfavorable for the actual application in the complex medium.
5. More characterization about SEM, TEM, HRTEM and XRD of the recovered gold particles are suggested to provide to facilitate readers' understanding.
6. Considering extremely high amount of Cu over Au, the energy changes of Cu need to be added

in DFT calculations.

7. The saturated adsorption capacity of this manuscript is 2.09 g/g. If the author carefully read more recent literatures, it can be found that adsorption performance of the proposed POPcarbene adsorbent is not superior in the field of gold recovery.

8. The preparation of Ptraiz-CN-A powder material is rather complicated and expensive, lack of superiority in the engineering application.

Reviewer #4 (Remarks to the Author):

In this study, it has been studied on the a poly(ionic liquid)-derived porous organic polycarbene (POPcarbene) adsorbent with superior gold-capturing capability. It has been also utilized theoretical calculations by Density Functional Theory. Study and its results are interesting and publishable after the authors should address the following comments.

- 1) Line 59 on page : Authors stated the reference of 15 as "Yavuz et al", however, that reference is not a reference by "Yavuz et al", so all references should be checked and corrected.
- 2) Las part of the Introduction section has some sentences about results/discussion/conclusion; this part should only have the aims of the study, so it should be corrected.
- 3) Some references should be given for the sentence located at line 127 on page 6.
- 4) Bigger geometries for the mechanism steps stated on Figure 5 should be given in supporting information.
- 5) Authors stated at line 30 on page 17 that "... in spite of the negligible energy barrier in step 2." However, no further information about on the transition state calculations, details for the TS calculations, geometry, activation barrier value, characterization of the TS geometry etc., These should be stated in the manuscript.
- 6) What are the spin multiplicity values for DFT calculations in Gaussian? If the spin multiplicity values greater than 1 (singlet) α and β molecular orbitals should be must be taken into attention.
- 7) Some error, called as Spin Contamination $\langle S^2 \rangle$, may be introduced into the calculations where spin multiplicity is utilized. The spin contamination value must be negligible (less than 10%, David C. Young, Computational Chemistry, 2001 John Wiley & Sons, Inc. page 228). Thus, related with (1), $\langle S^2 \rangle$ values should be given in text.
- 8) Spin Density values might be given for the atoms (especially gold atoms) on the structure. These values tell us where unpaired electrons are located in the system.
- 9) Some other critical values such as chemical hardness, chemical potential, electronegativity may calculated and used to comparison for activities These values can be easily calculated by using HOMO and/or LUMO values based on the approximation of Koopmans.
- 10) How did you characterized the geometries obtained by DFT calculations?
- 11) Vibrational Infrared frequencies can be calculated. They can be compared with the experimental values stated on page 7, and they can be used to characterize the geometries. Additionally, some mentions about the factor that will be used to scale the frequency values should be stated in text. (Frequency values should be scaled to reproduce experimental fundamentals, the factor and its reference(s) should be stated in text).
- 12) What are the convergence criteria in calculations? Gradients of root-mean-square (rms) displacement, max displacement, rms force, max force and the self-consistent field (SCF) convergence.
- 13) NBO analysis should be utilized on the geometries and charge values should be mentioned by using the experimental findings.
- 14) Some DFT calculations about solvent effect should be utilized and results should be compared with present values.
- 15) Do the energy values include Zero Point Energy (ZPE) corrections? If it does not contain, ZPE should be calculated and inserted into energy values or any comments on ZPE should be inserted into the related text
- 16) Energy values should be included thermal energy corrections.
- 17) Density-of-states and Partial Density-of-states and electronic configurations can be calculated in order to compare results.
- 18) Molecular Electrostatic Potential diagrams and HOMO and LUMO representations can be obtained and they can be compared with after/before adsorption.
- 19) Some comment for BSSE can be inserted into text.

Oct 26th, 2022

Subject: Response Letter

Dear Reviewers,

Thank you very much for your valuable comments on our manuscript entitled "Porous Organic Polycarbene Nanotrap for Efficient and Selective Gold Stripping from Electronic Waste". We greatly appreciate your critical suggestions. The manuscript has been carefully revised accordingly. All changes are highlighted in **Blue** in the manuscript for the convenience of reviewers. Our point-by-point responses to the comments are listed below. The manuscript after this revision process, in our opinion, has significantly improved its quality and readability, and meets the requirement of all reviewers. We hope that the revised manuscript would be now considered for publication in Nature Communications as an article.

Your Sincerely,

Yaozu Liao (on behalf of all coauthors)

Reviewer 1:

Li et al. reports gold capture by a network polymer made from a previously reported poly ionic liquid (Ptriaz, ref 23). The new polymer is made by treating Ptriaz with ammonia to make triazine crosslinkers. The solids were used in treating metal mixtures for selective metal uptake. The gold uptake reaches significant capacities of over 2 grams per gram polymer. Although the capacity and selectivity of Ptriaz towards gold is notable, there are major issues with the manuscript that need to be squarely addressed.

***Comment 1:** How is it possible to make a material hydrophilic after 11 min of water contact? Is it because there is some hydrolysis? Is it reversible? There should be some chemical explanation. (In 121)*

Reply: We thank Reviewer 1's valuable comments and suggestions. Through the unmodified drying process applied in the first trial, the Ptriaz-CN-A possessed limited porosities (177 m²/g) and exhibited a dense surface morphology under scanning electron microscopy (SEM) characterization, resulting in a large initial contact angle (109°). These conditions restrained the speed of water infiltration physically. Meanwhile, chemical process was also considered and evidenced by experiment. Proton nuclear magnetic resonance (¹H-NMR) was conducted for water soluble 1,2,4-triazolium monomer (TriazoleBr) in D₂O or DMSO-d₆ environment. As shown in **Fig. R1**, the signal of C5 proton (10.8 ppm) can be observed in DMSO-d₆ (the purple line in the bottom), while the C5 proton signal (10.4 ppm) gradually disappeared in D₂O solvent over time (the green, orange and red lines), which can be ascribed to the H/D exchange process. These results proved that C5-proton is highly active, and a reversible proton exchange take place between H₂O and Ptriaz-CN-A, resulting in a time-dependent interaction of Ptriaz-CN-A with water. Generally, the slow water infiltration of Ptriaz-CN-A can be attributed to its physical properties and reversible proton exchange without hydrolysis. The SEM characterization also revealed that the structure of the Ptriaz-CN-

A barely changed before and after water infiltration (**Fig. R2**).

After that, we further modified our supercritical CO₂ drying process and acquired P triaz-CN-A with a specific surface area of 332 m²/g (**Fig. R3**). Such material with higher porosities exhibited an even faster water infiltration compared to P triaz-CN-A with a relative lower surface area of 177 m²/g (6 min vs. 11 min), and the infiltration process is also reversible (**Fig. R4**). The results testify that the hydrophilicity of P triaz-CN-A could be easily achieved via a short time of water contact.

Revisions made, main text (Page 6 line 124): “**Fig. S5** shows that the surface layer has a water contact angle of about 60° initially, and after 6 min a spontaneous infiltration process occurred, indicating that the adsorbent in contact with water becomes superhydrophilic, and the infiltration process is reversible. The Scanning electron microscopy (SEM) characterization also revealed that the microstructure of the P triaz-CN-A barely changed before and after water infiltration (**Fig. S6**). Meanwhile, chemical process was also considered and evidenced by time-dependent ¹H-NMR test, the results proved that C5-proton is highly active, and a reversible proton exchange take place between H₂O and P triaz-CN-A, resulting in a time-dependent interaction of P triaz-CN-A with water (**Fig. S7**).”

Revisions made, supplementary information (Page 11 line 214): new **Fig. S7** was added in supplementary information.

Fig. R1 Time-dependent $^1\text{H-NMR}$ test of proton exchange of TriazoleBr. (Fig. S7 in revised supplementary information)

Revisions made, supplementary information (Page 10 line 211): new **Fig. S6** was added in supplementary information.

Fig. R2 The SEM images of PtriAZ-CN-A. (a) Original PtriAZ-CN adsorbent, (b) PtriAZ-CN adsorbent after water infiltration. (Fig. S6 in revised supplementary information)

Revisions made, main text (Page 5 line 97): new **Fig. 1c** was added in main text to replace previous Figure.

Fig. R3 N_2 sorption isotherms and pore size distribution (insets) for modified PtriAZ-CN adsorbent after supercritical CO_2 drying. (Fig. 1c in revised manuscript)

Revisions made, supplementary information (Page 10 line 208): new Fig. S5 was added in supplementary information to replace previous figure.

Fig. R4 Wetting process and water contact angle measurement of the modified P triaz-CN-A in cycle experiment. (**Fig. S5** in revised supplementary information)

***Comment 2:** The spectroscopic evidence for triazine formation is not adequate. It could simply be amidines that are forming, since ammonia reacting with nitrile normally produces amidine. Judging from the chelating activity of the adsorbent, it makes more sense that the structures are predominantly amidine networks. Low solubility is expected since amidines will have significant h-bonding. Also, the authors previous work (ref 23) employs ammonia to make membranes without any claim for triazine formation. This is confusing.*

Reply: We thank Reviewer 1's valuable comments and suggestions. We took the unpolymerized precursor-the TriazoleTFSI monomer to provide sufficient spectroscopic evidence for triazine/amidine formation. The Fourier transform infrared spectra (FT-IR) of the pristine TriazoleTFSI and NH_3 treated TriazoleTFSI (termed as TriazoleTFSI- NH_3) are shown in **Fig. R5a**. Compared with untreated TriazoleTFSI, the appearance of new absorption bands located at 1672 cm^{-1} (C=N) and 1225 cm^{-1} (C-N) support the cyclization reaction of nitrile groups into *s*-triazine ring (*ACS Macro. Lett.* 6, 1-5 (2017).). In addition, the TriazoleTFSI- NH_3 system also exhibits the stretching vibration modes of C=N (1610 cm^{-1}) for amidine, and 1642 cm^{-1} for protonated amidine (*Chem. Mater.* 22, 5492-5499

(2010).). Furthermore, the protonated amidine may arise because the H in the amidine will be removed by the excess NH_3 in the solution. To further investigate the chemical structure of samples, the ^{13}C NMR spectra were measured for typical TriazoleTFSI with or without ammonia treatment. As shown in **Fig. R5b**, carbon signals at 163 ppm (*Chem. Mater.* 22, 5492-5499 (2010)), 166-169 ppm (*Macromolecules* 53, 10366-10374 (2020)) and 170 ppm (*Angew. Chem. Int. Ed.* 57, 8438-8442 (2018)) were observed for ammonia treated TriazoleTFSI- NH_3 , but absent in the pristine TriazoleTFSI, which belong to the protonated amidine, conventional amidine and *s*-triazine, respectively. The above results prove that both *s*-triazine and amidine structures will exist in the ammonia-treated TriazoleTFSI.

However, different from monomer TriazoleTFSI, in P^{13}\text{C} NMR spectra are not obvious and cannot be directly distinguished from other signals. In contrast, characteristic peak of *s*-triazine is obviously seen and can be easily pointed out (**Fig. R6**) (*ACS Macro. Lett.* 6, 1-5 (2017) & *Mater. Horizons* 7, 2683-2689 (2020)). These results provided evidence that due to the “cation-methylene-nitrile”-sequence, the polymer Pvia nitrile cyclization under ammonia treatment to form *s*-triazine structures that covalently cross link the polymer (*Mater. Horizons* 7, 2683-2689 (2020)). Meanwhile, according to the evidence we obtained from the tests done to the monomer TriazoleTFSI, the amidine structures in the crosslinked polymers after ammonia treatment also cannot be simply ruled out. As a consequence, we believe both *s*-triazine and amidine structures coexist in the Ps-triazine structure may be the dominant in the polymer product. Hence, we revised the descriptions and schemes in the manuscript to clarify Reviewer 1’s concern.

Also, our previous work (ref 23) employs ammonia to make membranes without any claim for *s*-triazine formation. This is because the mechanisms behind the porous membrane formation were complex, involving both ionic crosslinks (obvious to us from the right beginning) and the *s*-triazine formation (not easy to identify, as nitrile’s trimerization

occurs usually under much harsher conditions). So, the crosslinking mechanisms were not fully revealed and neither thoroughly investigated (*Nat. Commun.* 9, 1717 (2018)). Two years after our work, one of our colleagues raised this question and clarified the unusual *s*-triazine formation mechanism very recently (*Mater. Horizons* 7, 2683-2689 (2020)). Based on what has been achieved, we believe that the “cation-methylene-nitrile” sequence in the PIL used would make crosslinking *via* nitrile cyclization under ammonia treatment to form *s*-triazine structures, similar to the paper published in *Mater. Horizons* 7, 2683-2689 (2020). Thus, both ionic and covalent crosslinking occur in our porous membrane systems.

[REDACTED]

Cited Fig. 1 Design of the “cation–methyl–nitrile” (CMN) sequence for the facilitated crosslinking of PILs. (*Mater. Horizons* 7, 2683-2689 (2020))

Fig. R6 (a) FT-IR and (b) ^{13}C NMR spectra of Ptriiaz-CN (bottom line) and the as-synthesized Ptriiaz-CN-A (top line).

Revisions made, main text (Page 7 line 142): “In order to further explore the chemical structure of crosslinked network, we first took unpolymerized precursor-the TriazoleTFSI monomer and and NH_3 treated TriazoleTFSI (termed as TriazoleTFSI- NH_3) to give sufficient spectroscopic evidence for the formation of networks. As shown in the Fourier transform infrared (FT-

IR) spectra of two substances (**Fig. S8a**), compared with untreated TriazoleTFSI, the appearance of new absorption bands located at 1672 cm^{-1} (C=N) and 1225 cm^{-1} (C-N) support the cyclization reaction of nitrile groups into *s*-triazine ring. In addition, the TriazoleTFSI-NH₃ monomer also exhibits the stretching vibration modes of C=N (1610 cm^{-1}) for amidine, and 1642 cm^{-1} for protonated amidine²⁹. Furthermore, the ¹³C NMR spectra were measured for typical TriazoleTFSI with or without ammonia treatment. As shown in **Fig. S8b**, carbon signals at 163 ppm, 166-169 ppm and 170 ppm were observed for ammonia treated TriazoleTFSI-NH₃, but absent in the pristine TriazoleTFSI, which belong to the protonated amidine, conventional amidine and *s*-triazine, respectively²⁹⁻³¹. The above results prove that both *s*-triazine and amidine structures will exist in the ammonia-treated TriazoleTFSI. However, different from monomer TriazoleTFSI, in P13C NMR spectra are not obvious and cannot be directly distinguished from other signals. In contrast, characteristic peaks of *s*-triazine are obviously seen and can be easily pointed out (**Fig. 1e & Fig. 1f**)^{26, 32}. These results provided evidence that due to the “cation-methylene-nitrile”-sequence, the polymer Pvia nitrile cyclization under ammonia treatment to form *s*-triazine structures that covalently crosslink the polymer²⁶. Meanwhile, according to the evidence we obtained from tests done to the monomer TriazoleTFSI, the amidine structures in the crosslinked polymers after ammonia treatment also cannot be simply ruled out. As a consequence, we believe both *s*-triazine and amidine structures coexist in the Ps-triazine structure is dominant in the polymer product.

Revisions made, supplementary information (Page 11 line 216): new **Fig. S8** was added in supplementary information.

Fig. R5 (a) FT-IR and (b) ¹³C NMR spectra of unpolymerized precursor-TriazoleTFSI (bottom line) and NH₃ treated TriazoleTFSI (top line). (Fig. S8 in revised supplementary information)

Comment 3: If there's a reductive removal for gold, why is Pt adsorbed as much as 78 % of gold? Pt uptake is much more than expected. What mechanism is at play for Pt?

Reply: We thank Reviewer 1's valuable comments and suggestions. Most metal ions applied in the experiment exist as cations (e.g. Cu^{2+} , Ni^{2+}) in aqueous solutions, while Au and Pt mainly exist as AuCl_4^- and PtCl_4^{2-} in water. Since the as-synthesized P triaz-CN-A is a cationic adsorbent, it exhibits strong electrostatic interactions with anionic metal ions like AuCl_4^- and PtCl_4^{2-} . Meanwhile, PtCl_4^{2-} is more negatively charged comparing with AuCl_4^- and it consequently displayed stronger electrostatic interaction with the cationic triazolium rings. Besides, PtCl_4^{2-} and the C5 proton of P triaz-CN-A is a Lewis pair ($\text{H}\cdots\text{Pt}^{2+}$) and also exhibited strong interactions. Therefore, the Pt uptake is a bit more than expected.

Comment 4: Cost issue is mentioned for introducing new materials but the synthesis is not short and uses costly reagents. It is not clear whether authors are making any gains there. There needs to be a LCA and process calculations to claim lowest cost.

Reply: We thank Reviewer 1's valuable comments and suggestions. According to your suggestions, we did a thorough life cycle assessment and cost analysis for the production of P triaz-CN-A. And please see the **Additional Supplementary Information-Life Cycle Assessment and Cost Analysis** for detailed data and analysis. Furthermore, our data collection and analysis are considered on a laboratory scale. As shown in **Fig. R7**, the results indicate that there is much room to reduce the environmental impact and cost of P triaz-CN-A production when they are produced at scale, and show emerging potentials for industrial productions. Furthermore, the production cost for synthesis of 1 g P triaz-CN-A in Scale-1 and Scale-2 is around 117.0 CNY and 107.3 CNY, respectively. The value of the gold captured by 1 g of P triaz-CN-A was approximately 795.2 CNY (gold price: about 380.5 CNY/g), and the profit margin will continue to increase by regenerating P triaz-CN-A. Therefore, P triaz-CN-A contributes a green and sustainable method for gold extraction from e-waste solution.

Revisions made, main text (Page 19 line 401): new **Fig. 6a** and **Fig.6b** was added in main text.

Fig. R7 (a) Relative environmental impacts of 1 g P triaz-CN-A production at different preparation scales. The higher environmental impact in each category is normalized. **(b)** Production cost distribution of raw materials, electricity in each step for the synthesis of 1 g P triaz-CN-A at different preparation scales. (**Scale-1**: small dose feeding, data based on our current research; **Scale-2**: big dose feeding, data based on the maximum production scale of the laboratory.) (Fig. 6 in revised manuscript)

Revisions made, main text (Page 19 line 408): “LCA is a systematic tool for determining the environmental impact of a product or a process across its entire life cycle or a portion of its life cycle^{50, 51}. So, the environmental impacts of P triaz-CN-A production prepared at different scales in this research were analyzed using the “cradle-to-gate” LCA approach (The detailed data and analysis are summarized in **Additional Supplementary Information-Life Cycle Assessment and Cost Analysis**). **Fig. AS1** shows the system boundary for this LCA study and the whole data was at the lab-scale. As shown in **Fig. 6a**, it is clear that the P triaz-CN-A prepared at Scale-2 has much lower environmental impacts across all categories. This is due to the reason that equipment can be operated under full load conditions and resources can be maximized. As a consequence, there is much room to reduce the environmental impact of P triaz-CN-A production when they are produced on industrial scale.

Cost analysis of raw materials and electricity in P triaz-CN-A preparation process in Scale-1 and Scale-2 are presented in **Fig. 6b** (The detailed data

and analysis are summarized in Additional Supplementary Information). The production cost for synthesis of 1 g PtriAZ-CN-A in Scale-1 and Scale-2 is around 117.0 CNY and 107.3 CNY, respectively, and more than 95 % of which are spent on raw materials. Although this production cost seems high, it should be noted that the cost here is based on our laboratory data. It is clear that the cost preparation is reduced when we scale up production under laboratory conditions (Scale-2). Therefore, the preparation cost of this adsorbent can be greatly reduced in industrial production. Furthermore, the value of the gold captured by 1 g of PtriAZ-CN-A was approximately 795.2 CNY (gold price: about 380.5 CNY/g), and the profit margin will continue to increase by regenerating PtriAZ-CN-A. In general, PtriAZ-CN-A contributes a green and sustainable method for gold extraction from e-waste solutions.

Comment 5: *100 ppb is not an “ultra-trace concentration”.*

Reply: We apologize this inaccurate expression. And we changed “ultra-trace concentration” to “trace concentration” in the manuscript.

Revisions made, main text (Page 12 line 252): “Generally, PtriAZ-CN-A exhibits a high sensitivity towards Au^{3+} (aq.) even under trace concentrations.

Comment 6: *The porosity (BET area of $177 \text{ m}^2/\text{g}$) is reported without any isotherms or calculations. Figure 1c is not showing any N_2 isotherms. BET range and calculations must be disclosed.*

Reply: We thank Reviewer 1’s valuable comments and suggestions. We have improved the preparation process. Specifically, we extended the time of crosslinking & solvent exchange process from 2 hours to 4 hours, and in the process of supercritical CO_2 drying process, the liquid CO_2 cleaning time was extended to 10 h to completely remove the residual DMF in the PtriAZ-CN-A. After certain adjustment of preparation methods, the specific Brunauer-Emmett-Teller (BET) surface area of PtriAZ-CN-A, evaluated with N_2 adsorption isotherm obtained at 77 K, is calculated to be $332 \text{ m}^2 \text{ g}^{-1}$ according to BJH model (**Fig. R3**).

As for the reported BET area, we believe there is a misunderstanding. Here we marked the adsorption-desorption curve in N_2 isotherms in **Fig.**

R3. Typically, the P triaz-CN-A is cooled, under vacuum, to cryogenic temperature (using liquid nitrogen). Nitrogen gas (as a typical adsorbate) is dosed to the P triaz-CN-A in controlled increments. After each dose of nitrogen gas, the relative pressure (P/P_0) is allowed to equilibrate, and the volume of nitrogen adsorbed is determined. Generally, Brunauer–Emmett–Teller (BET) equation is used to calculate the surface area of solid or porous materials, the BET equation can be described mathematically as follow:

$$V = \frac{V_m C P}{(P_0 - P)[1 + (C - 1)(P/P_0)]}$$

where V is adsorbed gas volume, P_0 is saturation pressure of adsorbate, P is equilibrium pressure of adsorbate, C is BET constant and V_m is the monolayer adsorbed gas volume.

As for the porosity (BET area of 332 m²/g) of P triaz-CN-A, it was calculated by BET model, which described the quantity of adsorbed gas as a function of the relative pressure. Moreover, the detailed information about isotherms and BET range reports, which are calculated and exported by Micromeritics software, are shown in **Fig. R8**.

Isotherm Tabular Report				
Relative Pressure (p/p ⁰)	Absolute Pressure (kPa)	Quantity Adsorbed (cm ³ /g STP)	Elapsed Time (h:min)	Saturation Pressure (kPa)
0.000210689	0.0215003	14.3075	03:44	102.0040538
0.001782076	0.1818090	27.4425	04:19	102.0476698
0.005523635	0.5634588	37.2122	04:45	102.0208735
0.010247311	1.0452504	43.4035	05:07	102.0087000
0.030777648	3.1402065	56.1464	05:17	102.0024099
0.052460768	5.3522441	63.6034	05:25	102.0287993
0.095014172	9.6924933	73.4842	05:32	102.0237460
0.152807621	15.5881880	83.6175	05:38	102.0110273
0.223117552	22.7810607	94.0730	05:44	102.0118492
0.293740468	29.9667747	103.9529	05:50	102.0137915
0.362750330	37.0060717	113.5922	05:56	102.0178927
0.432526785	44.1235626	123.9320	06:01	102.0152669
0.502100244	51.2256657	135.3914	06:07	102.0134848
0.571449190	58.2984947	148.8224	06:14	102.0227858
0.640696956	65.3666528	165.3296	06:21	102.0186846
0.677962207	69.1702797	176.1341	06:29	102.0242912
0.712204245	72.6708960	187.9294	06:36	102.0267487
0.780575005	79.6494578	219.7055	06:44	102.0365949
0.815126540	83.1673903	243.2261	06:57	102.0394674
0.884649245	90.2728507	332.2270	07:09	102.0300362
0.925480414	94.4392036	468.2576	07:37	102.0434384
0.956127490	97.5385507	685.5389	08:27	102.0434384
0.997650389	101.8091314	772.2103	09:27	102.0141683
0.947587140	96.7013960	761.2260	09:57	102.0489067
0.918848989	93.7753257	744.4898	10:00	102.0501354
0.892909815	91.1627145	645.5171	10:11	102.0573858
0.820740401	83.8253523	299.2560	11:03	102.0962173
0.753911996	76.9938325	223.8045	12:09	102.1338200
0.668871162	68.2879058	180.5121	12:31	102.1257559
0.605500196	61.8404731	161.2585	12:49	102.1247957
0.524900784	53.6013660	142.9348	12:58	102.1211013
0.473979707	48.3962021	133.5423	13:10	102.1171384
0.403627565	41.2083516	121.6767	13:16	102.1060635
0.332058988	33.9032825	111.0111	13:23	102.0949885
0.262375652	26.7879928	101.0114	13:29	102.1001802
0.192413652	19.6461794	90.4338	13:35	102.0978610
0.124552689	12.7172270	79.6398	13:41	102.1038745
0.086471037	8.8284010	72.4796	13:48	102.1031910
0.051080104	5.2146872	63.8973	13:54	102.0966241
			14:01	102.0884217

BET Report		
Relative Pressure (p/p ⁰)	Quantity Adsorbed (cm ³ /g STP)	1/[Q(p ⁰ /p - 1)]
0.052460768	63.6034	0.000870
0.095014172	73.4842	0.001429
0.152807621	83.6175	0.002157
0.223117552	94.0730	0.003053
0.293740468	103.9529	0.004001

BET Surface Area: 332.0488 ± 2.1989 m²/g
Slope: 0.012917 ± 0.000085 g/cm³ STP
Y-Intercept: 0.000191 ± 0.000016 g/cm³ STP
C: 68.634798
Qm: 76.2878 cm³/g STP
Correlation Coefficient: 0.9999345
Molecular Cross-Sectional Area: 0.1620 nm²

Fig. R8 The detailed information about Isotherm Tabular Report and BET Report.

Revisions made, main text (Page 7 line 133): “Hence, N₂ sorption (at 77 K) measurement was performed to access the pore characteristics of the P2 g⁻¹ (**Fig. 1c**), which is fairly acceptable for porous materials prepared free of external templates.”

Revisions made, main text (Page 5 line 97): new **Fig. 1c** was added in main text to replace previous Figure.

Fig. R3 N₂ sorption isotherms and pore size distribution (insets) for modified P2 drying. (**Fig. 1c** in revised manuscript)

Formatting:

Comment 1: Typos: *has been comined, traizolium, non-prescious metals,*

Reply: We have carefully checked whole the manuscript and all revised mentioned typos have been revised in the manuscript accordingly.

Revisions made, main text (Page 4 line 77): “has been comined” was corrected into “has been combined”.

Revisions made, main text (Page 5 line 107): “traizolium-based” was corrected into “triazolium-based”.

Revisions made, main text (Page 12 line 249): “non-prescious metals” was corrected into “non-precious metals”.

Comment 2: Ptraiz-CN vs Ptriaz-CN. Please make sure you're consistent with codes.

Reply: We thank Reviewer 1's valuable comments and suggestions. We have checked our manuscript thoroughly and revised mentioned "Ptraiz-CN" into "Ptriaz-CN".

Revisions made, main text: All "Ptraiz-CN" were corrected into "Ptriaz-CN".

Reviewer 2:

In general, although authors are highly recommended to conduct some extra tests for more clarifications (which is mentioned in the following pages), the manuscript worth publishing after major revisions.

Comment 1: Despite "Conclusion" which is well-written, "Abstract" needs further rewriting to make it easier to grab the novel ideas of the research.

Reply: We thank Reviewer 2's valuable comments and suggestions. We have revised "Abstract" in the manuscript accordingly.

Revisions made, main text (Page 2 line 23): "N-heterocyclic Carbenes, as one traditional reactive or binding sites, have been long studied in chemical catalysis. However, their unique properties had barely been applied in functional materials. Herein, we report the design and preparation of a poly(ionic liquid)-derived porous organic polycarbene (POPcarbene) adsorbent with superior gold-capturing capability. With carbene site in the porous network as the "nanotrap", it exhibits an ultrahigh gold recovery capacity of 2.09 g/g and outstanding concentrating power for gold ion in its aqueous solution even at 100 ppb level, through a reduction-promoted adsorption process. In-depth exploration in a complex metal ion environment (Au^{3+} , Pt^{2+} , Cu^{2+} , Ni^{2+} , etc.) of an electronic waste-extraction solution proved POPcarbene adsorbent with a significant gold recovery efficiency (REE) of 99.8 %. X-ray photoelectron spectroscopy (XPS) study along with nuclear magnetic resonance (NMR) spectroscopy reveals that the high performance of the POPcarbene adsorbent results from the formation of robust metal-carbene bond in the porous polycarbene nanotraps plus the power to reduce close-by gold ions into nanoparticles. Density functional theory (DFT) calculations indicate the energetically favorable multinuclear-Au binding enhances adsorption as clusters, leading to a surprisingly high gold capacity, where the capture abilities for metals other than Au (Pt, Cu, etc.) stay predominantly at electrostatic interactions. Life cycle assessment (LCA) and cost analysis indicate that

the synthesis of POPcarbene adsorbent meets the criteria for green chemistry principles and shows emerging potentials for industrial productions. These results reveal the potentials to apply "carbene chemistry" into materials science and highlight POPcarbene as rising materials for precious metal recovery, who may guide their future exploration strategies for real-life implications."

Comment 2: Introduction is well-written.

Reply: We thank Reviewer 2's encouraged comments.

Comment 3: Line 151. This method is not HADDF. This is STEM or qualitative EDS and should not be mixed up with other advanced methods.

Reply: We thank Reviewer 2 pointed out this inaccurate expression. We have changed "HADDF-STEM images" to "STEM images" in the manuscript.

Revisions made, main text (Page 9 line 175): "HADDF-STEM images" was corrected into "STEM images".

Comment 4: Line 153. How did you analyze particle size of Au? From BET? when there are a porous material and particles next to each other in BET analysis, particle size and porosity size are mixed up and not reliable. Any other method was employed?

Reply: We thank Reviewer 2's valuable comments. Indeed, we used a software called ImageJ to analyze the particle size of Au particles from STEM images. Specifically, we marked more than 100 Au nanoparticles in the STEM images for Au-loaded P-CN-A, and then used the software-ImageJ to make statistics on the particle size of Au nanoparticles (**Fig. R9**).

Revisions made, main text (Page 10 line 221): "...is further verified by gold nanoparticles of 15-30 nm in size observed in the TEM images (**Fig. 2e, 2f** and **Fig. S13**) ..."

Revisions made, supplementary information (Page 14 line 228): new **Fig. S13** was added in supplementary information.

Fig. R9 Statistics on the Au particle size of Au-loaded Ptriz-CN-A by ImageJ software. (Fig. S13 in revised supplementary information)

Comment 5: Line 158. I think 24 h is a quite a long time. Usually adsorption takes places within a few hours. Such a long time is a serious disadvantage in industrial scale, particularly since it needs stirring that consumes energy. However, regarding Table S2, it is acceptable for research purpose.

Reply: We thank Reviewer 2's valuable comments. In the adsorption-isotherm experiment, different concentrated metal ion solutions need to be placed in containers, together with adsorbents, and stirred for adsorption test. After reaching the adsorption equilibrium, the concentration of metal ions (C_e) can be measured and applied to calculate equilibrium adsorption quantity (Q_e). This experiment needs to ensure that the stirring time for each sample is identical. In order to reach adsorption equilibrium in different concentrated Au^{3+} solutions, we set the stirring time as 24 h by referring to previous reports (*J. Am. Chem. Soc.* 140, 16697-16703 (2018) and *ACS Appl. Mater. Interfaces* 14, 11803-11812 (2022)). Meanwhile, we also tested the adsorbent when the initial concentration of Au^{3+} solution is 100 ppm, the adsorption equilibrium can be reached **within 30 min**.

Comment 6: Line 162. The fitting for both Pt and Au are not very strong (0.89 and 0.87) but acceptable. it means you can look for other models to fit on your data.

Reply: We thank Reviewer 2's valuable comments and suggestions. In addition to Freundlich adsorption isotherm model, the Langmuir adsorption isotherm model is commonly used in adsorption experiments to fit data. The Langmuir adsorption isotherm model assumes that all adsorption sites in the adsorbent are equivalent, which forms a homogeneous surface with the same force on the surface of the adsorbent. The Langmuir adsorption isotherm model can be described mathematically as follows:

$$\frac{C_e}{Q_e} = \frac{1}{Q_{max} \times k_L} + \frac{C_e}{Q_{max}}$$

Where C_e is the equilibrium concentration (mg g^{-1}), Q_e is the equilibrium adsorption capacity (mg g^{-1}), Q_{max} is the maximum adsorption amount at the time of equilibrium (mg g^{-1}), k_L (L mg^{-1}) is the Langmuir adsorption isotherm constant.

According to Reviewer 2's suggestions, we made new adsorption isotherm curves of Au^{3+} and Pt^{2+} adsorption by P triaz-CN-A, fitted by Langmuir model. They are shown in the **Fig. R10** and **Fig. R11**. As can be seen in the figure, the isotherm fittings are largely improved for Au^{3+} .

Revisions made, main text (Page 10 line 184): “**Fig. 2a** and **Fig. S9** show the adsorption isotherms of Au^{3+} and Pt^{2+} on the P triaz-CN-A. It can be seen that the adsorption amounts increase with the increasing $\text{Au}^{3+}/\text{Pt}^{2+}$ concentration, and the adsorption capacity could greatly increase and reach its maximum at high concentrations. The P triaz-CN-A seized significant amounts of gold ions with an uptake capacity of as high as 2.09 g/g, which outperforms most reported porous organic polymers (see **Table S1**). By contrast, in the same experiment, the adsorption amount of platinum in P triaz-CN-A is 1.62 g/g, nearly 78 % of the gold case. Moreover, the Freundlich adsorption isotherm model and Langmuir adsorption isotherm model were employed to analyze the adsorption isotherm data. The results indicated that the adsorption isotherms were well-fitted with the Langmuir model (**Fig. 2a, Fig. S10 & Fig. S11**), which yielding the linear correlation

coefficients as high as 0.988 for Au^{3+} and 0.88 for Pt^{2+} , respectively.

Revisions made, main text (Page 9 line 171): new **Fig. 2a** was added in main text to replace previous Figure.

Fig. R10 Adsorption isotherm of Au^{3+} on PtriAZ-CN-A and related Langmuir isotherm fitting ($R^2=0.988$). (**Fig. 2a** in revised manuscript).

Revisions made, supplementary information (Page 12 line 219): new **Fig. S9** was added in supplementary information.

Fig. R11. Adsorption isotherm of Pt^{2+} on PtriAZ-CN-A and related Langmuir isotherm fitting ($R^2=0.880$). (**Fig. S9** in revised supplementary information).

Revisions made, supplementary information (Page 3 line 66): “The Langmuir adsorption isotherm model assumes that all adsorption sites in the adsorbent are equivalent, which forms a homogeneous surface with the

same force on the surface of the adsorbent. The Langmuir adsorption isotherm model can be described mathematically as follows:

$$\frac{C_e}{Q_e} = \frac{1}{Q_{max} \times k_L} + \frac{C_e}{Q_{max}}$$

Where C_e is the equilibrium concentration (mg g^{-1}), Q_e is the equilibrium adsorption capacity (mg g^{-1}), Q_{max} is the maximum adsorption amount at the time of equilibrium (mg g^{-1}), k_L (L mg^{-1}) is the Langmuir adsorption isotherm constant.”

Comment 7: Line 165. Does it mean that the adsorbent cannot selectively separate Pt group metals (PGMs) and Au? how about Ag?

Reply: We thank Reviewer 2’s valuable comments and suggestions. In accordance with Reviewer 2’s concerns, the P triaz-CN-A cannot selectively separate Pt^{2+} and Au^{3+} with high efficiency. When the initial concentration of the two metal ions is 5ppm, the recovery efficiency of the P triaz-CN-A for Au^{3+} and Pt^{2+} is 99.77 % and 97.91 %, respectively, after stirring for 6 hours (**Fig. R12**). Since the as-synthesized P triaz-CN-A is a cationic adsorbent, it exhibits strong electrostatic interactions with anionic metal ions like AuCl_4^- and PtCl_4^{2-} . Meanwhile, PtCl_4^{2-} is more negatively charged comparing with AuCl_4^- and it consequently displayed stronger electrostatic interaction with the cationic triazolium rings. Besides, PtCl_4^{2-} and the C5 proton of P triaz-CN-A is a Lewis pair ($\text{H} \cdots \text{Pt}^{2+}$) and also exhibited strong interactions.

To investigate the interaction and adsorption ability of P triaz-CN-A towards silver ions, the adsorption isotherm experiment was performed. Typically, 5 mg of P triaz-CN-A was placed in 10 mL aqueous solutions with varying Ag^+ concentrations (100-900 ppm). The solutions were stirred for 24 h at 600 rpm under dark conditions to achieve adsorption equilibrium. The solutions were filtered through a 0.45 μm syringe filter units and the filtrate was analyzed *via* ICP-OES to determine the residual Ag^+ concentrations. **Fig. R13** showed the adsorption isotherm of Ag^+ on P triaz-CN-A, and the adsorption isotherm data was analyzed in detail by the Langmuir adsorption isotherm model. It can be seen that the adsorption amount increases with the increasing Ag^+ concentration, and the adsorption

capacity reaches its maximum at high concentration and achieves adsorption equilibrium. *The adsorption amount of silver in PtriAZ-CN-A is approximately 495 mg/g, far less than Au and Pt.* The above results proved that the interaction between PtriAZ-CN-A and silver was weak, resulting in a very low adsorption capacity.

For the selective adsorption of Au^{3+} / Pt^{2+} / Ag^+ by the PtriAZ-CN-A, since the metal salts of gold and platinum we selected are HAuCl_4 and K_2PtCl_4 , and the metal salt of silver is generally AgNO_3 , but the mixed aqueous solution of these three metal salts will produce AgCl precipitation, so subsequent selective adsorption experiments cannot be performed to determine the selective adsorption efficiency of the PtriAZ-CN-A between Au^{3+} , Pt^{2+} and Ag^+ .

Fig. R12 Selective adsorption of Au^{3+} / Pt^{2+} on PtriAZ-CN-A.

Fig. R13 Adsorption isotherm of $\text{Ag}(\text{I})$ on PtriAZ-CN-A and related Langmuir isotherm fitting ($R^2=0.998$).

Comment 8: Line 167. Concentration of Au and Pt in Table S1 is more or less acceptable for e-waste treatment; however, concentration of other metals particularly Cu and Ni are usually far more (in traditional, cheap, and more efficient recovery methods) and are in the range g/l. So, you need to duplicate the tests using solution with high concentration of Cu and Ni too that are close to the real e-waste solutions.

Reply: We thank the Reviewer 2's valuable comments and suggestions.

Firstly, we duplicated the test to get the e-waste solution. The detailed information about concentration of metals is shown in **Table. R1**. The results showed that there was little difference between the concentration of metal ions obtained after the duplicated leaching test. We think this can be attributed to the leaching method we chose. As mentioned in the manuscript, we used Yang's NBS/Py method to extract gold from electronic waste (*Angew. Int. Ed.* 56, 9331-9335 (2017).). Compared with other traditional leaching method (e.g. using aqua regia to leach gold from e-waste, and this yields an extremely acidic gold-containing leachate), this NBS/Py method has been proved to exhibit significant Au leaching preference over other traditional cheap metals (e.g. Cu, Ni, Mg, Zn and so on). This is why the content of Cu and Ni in our electronic wastewater is not very high. And the concentrations of metal ions in the e-waste water obtained in our experiments are in consistent with the other reports (*ACS Sustain. Chem. Eng.* 10(30), 9719-9731 (2022)., *Chem. Mater.* 32, 5343-5349 (2020).), which also used this NBS/Py method to get leaching solution. Based on the above discussion, we think this state-of-the-art leaching method is the best choice, and the high leaching selectivity for Au easily afforded the straightforward adsorption for Au³⁺ in e-waste solution with high efficiency.

And in Line 167 (previous manuscript), we believe there is a misunderstanding. Here we set the initial concentration of Au³⁺ solution to 100 ppm not to carry out the gold extraction experiment of authentic electronic wastewater or the selective adsorption experiment of metal ions, but to carry out the adsorption kinetic experiment, and study the influence of solutions' pH on the adsorption process. The adsorption kinetics experiments of P triaz-CN-A were conducted by collecting samples at

different time intervals. The adsorbent was then filtered by 0.45 μm syringe filter units and the remaining Au^{3+} concentrations in solution were determined by ICP-OES, so as to calculate the removal rate of Au^{3+} by the adsorbent. Commonly, most studies (*ACS Sustain. Chem. Eng.* 10(30), 9719-9731 (2022)., *ACS Appl. Mater. Interfaces* 12, 30474-30482 (2020)., *ACS Appl. Mater. Interfaces* 14, 11803-11812 (2022). *Chem. Eng. J.* 410, 128360 (2021).) also set the initial concentration of metal ion solution as 100 ppm.

Revisions made, supplementary information (Page 25 line 294): new **Table. S4** was added in supplementary information.

Table. R1 Metal content of authentic electronic wastewater (duplicated test). (**Table. S4** in revised supplementary information)

Element	Test-1	Test-2
	Concentration (ppm)	Concentration (ppm)
Cu	346.293	357.56
Ni	58.31	85.32
Pt	5.765	0.534
Au	1.585	2.235
Zn	1.552	1.832

Comment 9: Line 171-173. Please describe why pH change is not effective? Functional groups are effective?

Reply: We thank the Reviewer 2's valuable comments. In the cationic 1,2,4-triazolium ring, the C5 proton is highly active and undergoes easier deprotonation. In an aqueous solution, the proton exchange process between P_{triaz}-CN adsorbent and H₂O continues constantly. When the pH of the solution is 2, 4 and 7, the proton exchange process will not be affected, and no other negative ions in the solution can compete with AuCl_4^- for electrostatic interactions with P_{triaz}-CN adsorbent. So, when the pH is below 7, the adsorption process of P_{triaz}-CN adsorbent on AuCl_4^- is not affected by the pH of the solution. When the pH is above 7, the excessive OH^- is prone to compete for the adsorption sites, thus, the recovery efficiency of P_{triaz}-CN-A on AuCl_4^- is affected slightly. However,

the properties of Ptri az-CN-A in alkaline solution do not change, and Ptri az-CN-A is still a cationic adsorbent, could also adsorb AuCl_4^- by electrostatic interactions.

Revisions made, main text (Page 10 line 198): “The adsorption behavior of Ptri az-CN-A exhibits weak correlation with the pH value of treated solutions, since the Ptri az-CN-A is a cationic adsorbent, and its properties (electrostatic interactions with AuCl_4^-) will not change in acidic or alkaline solutions.

Comment 10: Line 201. As mentioned, 5 ppm is acceptable for Pt and Au but not at all for Cu, Ni or Co. Other metals are usually exist in the order of g/l in the e-waste leaching solution. Please duplicate the test in high concentration of Cu and Ni.

Reply: We thank the Reviewer 2’s valuable comments and suggestions. Ni^{2+} and Cu^{2+} are the most common coexisting elements with Au^{3+} in e-waste leaching solution and may interfere with the adsorption of Au^{3+} . To simulate more realistic electronic wastewater, we prepared e-waste solutions contain multiple interfering ions including Co^{2+} , Zn^{2+} , Cu^{2+} , Ni^{2+} , Mg^{2+} , and Cr^{2+} with their concentration at 300 ppm, which is 60 times higher than that of target Au^{3+} and Pt^{2+} . The **Fig. R14** shows the selective adsorption of Au^{3+} on Ptri az-CN-A in the presence of other highly concentrated metal ions in aqueous solution. As a result, Ptri az-CN-A exhibits nearly a full Au^{3+} recovery (REE of 99.8 %), leaving a Au^{3+} concentration down to 10 ppb. A similar adsorption preference was found in another precious metal ion Pt^{2+} with a REE of up to 98.9 %. In case of other metal ions like Cr^{2+} , though the REE of Ptri az-CN-A for Cr^{2+} is 46.1 %, the original concentration of Cr^{2+} in electronic wastewater is much less in real case. Moreover, the REE of Ptri az-CN-A for Mg^{2+} and Co^{2+} are 21.5 % and 1.75 %, respectively. As for major metal ions (Cu^{2+} and Ni^{2+}) in e-waste solution, the Ptri az-CN-A showed weak adsorption effect on Cu^{2+} and Ni^{2+} , with REE of 4.53 % and 18.6 %, respectively, in spite of their 60 times higher concentrations comparing with that of Au^{3+} . The results suggested a remarkable anti-interference ability of Ptri az-CN-A for Au^{3+} adsorption.

Revisions made, main text (Page 11 line 230): “To simulate more realistic electronic wastewater, we prepared e-waste solutions contain multiple interfering ions including Co^{2+} , Zn^{2+} , Cu^{2+} , Ni^{2+} , Mg^{2+} , and Cr^{2+} with their concentration at 300 ppm, which is 60 times higher than that of target Au^{3+} and Pt^{2+} . Typically, 5 mg of the Ptri az-CN-A was placed in a 10 mL mixture metal ion solution, and after 6 h, their remaining concentrations in the solution were measured by ICP-OES. The **Fig. 2g** shows the selective adsorption of Au^{3+} on Ptri az-CN-A in the presence of other highly concentrated metal ions in aqueous solution. As a result, Ptri az-CN-A exhibits nearly a full Au^{3+} recovery (REE of 99.8 %), leaving a Au^{3+} concentration down to 10 ppb. A similar adsorption preference was found in another precious metal ion Pt^{2+} with a REE of up to 98.9 %. In case of other metal ions like Cr^{2+} , though the REE of Ptri az-CN-A for Cr^{2+} is 46.1 %, the original concentration of Cr^{2+} in electronic wastewater is much less in real case. Moreover, the REE of Ptri az-CN-A for Mg^{2+} and Co^{2+} are 21.5 % and 1.75 %, respectively. As for major metal ions (Cu^{2+} and Ni^{2+}) in e-waste solution, the Ptri az-CN-A showed weak adsorption effect on Cu^{2+} and Ni^{2+} , with REE of 4.53 % and 18.6 %, respectively, in spite of their 60 times higher concentrations comparing with that of Au^{3+} . The results suggested a remarkable anti-interference ability of Ptri az-CN-A for Au^{3+} adsorption.

Revisions made, main text (Page 9 line 171): new Fig. 2g was added in main text to replace previous Figure.

Fig. R14 Selective adsorption of Au^{3+} on Ptriz-CN-A in simulated electronic wastewater. (**Fig. 2g** in revised manuscript).

Comment 11: Line 208. This K_d is only valid once the concentrations of other ions are identical to Au. Need to increase make a more realistic e-waste solution.

Reply: We thank Reviewer 2’s valuable comments and suggestions. To simulate more realistic electronic wastewater, the prepared e-waste solution contains multiple interfering ions including Co^{2+} , Zn^{2+} , Cu^{2+} , Ni^{2+} , Mg^{2+} , and Cr^{2+} with their concentration at 300 ppm, and the concentrations of Au^{3+} and Pt^{2+} are 5 ppm. The affinity of P triaz-CN-A for Au^{3+} is quantified by comparing the distribution coefficients (K_d , mL/g) of different ions. As can be seen from **Table. R2**, P triaz-CN-A has the largest distribution coefficient for Au^{3+} , and the K_d -based ion selectivity coefficient is ~ 6 for $\text{Au}^{3+}/\text{Pt}^{2+}$, 2.2×10^3 for $\text{Au}^{3+}/\text{Ni}^{2+}$, 1.1×10^4 for $\text{Au}^{3+}/\text{Cu}^{2+}$, 2.8×10^4 for $\text{Au}^{3+}/\text{Co}^{2+}$, 1.1×10^4 for $\text{Au}^{3+}/\text{Zn}^{2+}$, 0.6×10^3 for $\text{Au}^{3+}/\text{Cr}^{2+}$, and 1.8×10^3 for $\text{Au}^{3+}/\text{Mg}^{2+}$. This data proves the preferential uptake of P triaz-CN-A to noble metal ions (Au^{3+} and Pt^{2+}) over non-precious metals in the e-waste solutions.

Revisions made, main text (Page 12 line 245): “As can be seen from **Table S3**, P triaz-CN-A has the largest distribution coefficient for Au^{3+} , and the K_d -based ion selectivity coefficient is ~ 6 for $\text{Au}^{3+}/\text{Pt}^{2+}$, 2.2×10^3 for $\text{Au}^{3+}/\text{Ni}^{2+}$, 1.1×10^4 for $\text{Au}^{3+}/\text{Cu}^{2+}$, 2.8×10^4 for $\text{Au}^{3+}/\text{Co}^{2+}$, 1.1×10^4 for $\text{Au}^{3+}/\text{Zn}^{2+}$, 0.6×10^3 for $\text{Au}^{3+}/\text{Cr}^{2+}$, and 1.8×10^3 for $\text{Au}^{3+}/\text{Mg}^{2+}$. This data proves the preferential uptake of P triaz-CN-A to noble metal ions (Au^{3+} and Pt^{2+}) over non-precious metals in the e-waste solutions.”

Revisions made, supplementary information (Page 25 line 293): new **Table. S3** was added in supplementary information.

Table R2. The K_d values of multiple metal ions in simulated e-waste solution. (**Table. S3** in revised supplementary information)

	Coexisting Ions							
K_d	Au^{3+}	Pt^{2+}	Cu^{2+}	Ni^{2+}	Zn^{2+}	Mg^{2+}	Co^{2+}	Cr^{2+}
	1011.582	174.772	0.0948	0.4564	0.0882	0.548	0.0357	1.712

Comment 12: Line 218. These pins are usually coated by Au. So you need to cut the pin and then analyses it and provide the results too.

Reply: Thanks for Reviewer 2’s valuable comments, we have made a detailed analysis of the CPU pins according to your suggestion. The EDX analysis and SEM mapping (Fig. R15) showed that the CPU pins were mainly composed of inner Cu cores (> 99.0 wt%) and outer Au coatings (> 95.0 wt%). In addition, EDX analysis evidenced the presence of Ni (> 4.0 wt%) in the CPU pins, which mainly served as a middle layer between Au and Cu to promote the stable adhesion of Au to the Cu core (*Appl. Surf. Sci.* 185, 289 (2002).), and there are also trace amounts of Pt (0.24 wt%) in the inner cores.

Revisions made, main text (Page 13 line 264): “X-ray spectroscopy (EDS) mapping for one single pin in the scrap CPU (Fig. 3a & Fig. S16).”

Revisions made, supplementary information (Page 15 line 236): new Fig. S16 was added in supplementary information.

Fig. R15 The corresponding EDX and SEM mapping spectra showing the elemental composition on the point of pin (A), cross section of pin (B) and surface of pin (C). (Fig. S16 in revised supplementary information).

Comment 13: Line 228. Are N-bromosuccinimide (NBS) and pyridine (Py) sustainable chemicals? I believe the environmental impact of such chemicals might be more than traditional methods. It needs more justification to use such chemicals.

Reply: Thanks for Reviewer 2’s valuable comments. We apologize for the

misunderstanding raised here, the mentioned leaching route in our manuscript is not firstly proposed and applied, this is a commonly-used and environment-friendly protocol for highly efficient Au extraction (*Angew. Int. Ed.* 56, 9331-9335 (2017)). Moreover, there are many other studies that have used this approach to directly leach Au⁰ from electronic waste to form Au³⁺ in water.

Compared with traditional alkali cyanide leaching method, who exhibited few obvious drawbacks (*e.g.*, its well-known lethal toxicity, risky explosiveness, high energy consumption, *etc.*), this NBS/Py method shows significant Au leaching preference over other metals and achieves an approximately 90 % leaching efficiency of Au at room temperature with a nearly neutral pH. Moreover, the minimum dose of NBS/Py is as low as 10 mM, which exhibits low toxicity towards aquatic creatures. In our research, the metal leaching solution was prepared by mixing 0.966 μ L of pyridine and 750 mg of NBS in 120 mL of DI water. 0.21 g of the metal scraps were put into the solution and the mixture was let stand for 4 days.

[REDACTED]

Cited Fig.2 The proposed NBS/Py method to leach gold from ores and WEEE (*Angew. Int. Ed.* 56, 9331-9335 (2017)). **Note: the fish in solution is still alive.**

Generally, this strategy shows low environmental impact with negligible cytotoxicity. This NBS/Py method is important as it provides a simple, eco-friendly, feasible option to leach gold, by reducing the total chemical waste and energy load. That's why we chose it.

Revisions made, main text (Page 13 line 267): “Yang’s leaching route employs mild and environmentally benign conditions (neutral pH, and chemicals of low toxicity) that can efficiently oxidize Au⁰ in electronic waste into Au³⁺ in a high yield and selectivity.”

Comment 14: Line 232. Please correct it to: "... was filtered and then acidified by HCl to pH = 2 ..."

Reply: Thanks for Reviewer 2’s valuable comments and suggestions. As you mentioned, we have corrected it to: “... was filtered and then acidified by HCl to pH = 2 ...” in our revised manuscript.

Revisions made, main text (Page 14 line 270): “...the resulting e-waste solution was filtered and then acidified by HCl to pH = 2...”

Comment 15: Line 237. Please repeat the test with high concentration of Cu and Ni, since usually in e-waste leaching solutions, Cu and Ni are in high concentrations.

Reply: Thanks for Reviewer 2’s valuable comments and suggestions. As you mentioned in the authentic e-waste solution (Comment 8), we have duplicated the test again to get the e-waste solution. The detailed information about concentration of metal ions in e-waste leaching solution is shown in the **Table. R1** The results showed that there was little difference between the concentration of metals obtained after the duplicated leaching test. We think this is attributed to the leaching method we chose. As mentioned in the manuscript, we used Yang's NBS/Py method to extract gold from electronic waste (*Angew. Int. Ed.* 56, 9331-9335 (2017).). Compared with other traditional leaching method (e.g., using aqua regia to leach gold from e-waste, and this yields an extremely acidic gold-containing leachate), this NBS/Py method exhibited significant Au leaching preference over other traditional cheap metals (e.g. Cu, Ni and Zn). This is why the content of Cu and Ni in our e-waste solution is not very high. And the concentrations of metal ions in the e-waste water obtained in our experiments are in consistent with the other studies (*ACS Sustain. Chem. Eng.* 10(30), 9719-9731 (2022)., *Chem. Mater.* 32, 5343-5349 (2020).), which also used this NBS/Py method to get leaching solutions. Based on the above discussion, we think this state-of-the-art

leaching method is the best choice, and the high leaching selectivity for Au can easily afford the straightforward adsorption for Au³⁺ in e-waste solution with high efficiency.

In such leaching solution, Cu²⁺ is the major metal component (357.56 ppm), and Au³⁺ is the minority (2.235 ppm). Still, P3+ (REE ~ 92.75 %) and Pt²⁺ (REE ~ 36.97 %) in the presence of abundant Cu²⁺ and other metal ions. By contrast, Cu²⁺ concentration in e-waste water is only reduced approximately by 8 ppm to 349.22 ppm, giving an ultralow REE of 2.33 % (**Fig. R16**).

Revisions made, supplementary information (Page 25 line 294): new Table. S4 was added in supplementary information.

Table. R1 Metal content of authentic electronic wastewater (duplicated test). (**Table S4** in revised supplementary information)

Element	Test-1	Test-2
	Concentration (ppm)	Concentration (ppm)
Cu	346.293	357.56
Ni	58.31	85.32
Pt	5.765	0.534
Au	1.585	2.235
Zn	1.552	1.832

Revisions made, main text (Page 13 line 254): new Fig. 3c was added in main text to replace previous Figure.

Fig. R16 Gold recovery efficiency from actual e-waste solution using PFig. 3c in revised manuscript).

Revisions made, main text (Page 13 line 272): “In such leaching solution, Cu^{2+} is the major metal component (357.56 ppm), and Au^{3+} is the minority (2.235 ppm). Still, PtriAZ-CN-A displayed a satisfied selectivity for Au^{3+} (REE ~ 92.75 %) and Pt^{2+} (REE ~ 36.97 %) in the presence of abundant Cu^{2+} and other metal ions. By contrast, Cu^{2+} concentration in e-waste water is only reduced approximately by 8 ppm to 349.22 ppm, giving an ultralow REE of 2.33 % (Fig. 3c).”

Comment 16: Line 247. How many cycle can the adsorbent last before reaching to 50 % of adsorption capacity?

Reply: Thanks for Reviewer 2’s valuable comments. The reusability of the material is of vitality for practical application. Multiple adsorption-desorption experiments were performed to evaluate PtriAZ-CN-A’s performance. **Fig. R17** shows the results of 11 cycles of the reusability of PtriAZ-CN-A for Au^{3+} recovery. The results demonstrate that the recovery efficiencies for the PtriAZ-CN-A were all above 95 % after six cycles, indicating its excellent reusability. The recovery efficiency decreased continuously in the subsequent cycles owing to the loss of the free spaces of adsorbents due to the continuous enrichment of the reduced Au nanoparticles in the skeleton of PtriAZ-CN-A. The recovery efficiency decreased to 44.13 % in the 10th cycle.

Revisions made, main text (Page 13 line 254): new **Fig. 3d** was added in main text to replace previous Figure.

Fig. R17 Regeneration and reusability of PtriAZ-CN-A. (**Fig. 3d** in revised manuscript).

Revisions made, main text (Page 14 line 290): “**Fig. 3d** shows the results of 11 cycles of the reusability of P-CN-A for Au³⁺ recovery. The results demonstrate that the recovery efficiencies for the P-CN-A were all above 95 % after six cycles, indicating its excellent reusability. The REE decreased continuously in the subsequent cycles owing to the loss of the free spaces of adsorbents due to the continuous enrichment of the reduced Au nanoparticles in the networks of P-CN-A. Furthermore, the regenerated P-CN-A still exhibits more than 60 % elution efficiency (EEE) for Au³⁺ after eleven adsorption-desorption cycles (**Fig. S21**).

Comment 17: Methods. *Fig. S17 has never been cited in the main text.*

Reply: Thanks for Reviewer 2’s valuable comments. We removed the previous Fig. S17 in the revised supplementary information.

Reviewer 3:

This manuscript reports the design and preparation of polyionic liquid-derived porous organic polycarbene (POPcarbene) adsorbent and investigated its gold recovery performance from e-waste.

***Comment 1:** Although some results have been achieved, authors only do much routine work to follow the findings of previous reports, such as POP-pNH₂-py (*Angew. Chem. Int. Ed.* 2020, 59, 1-6) and Triazine-based POPs (*Chemical Engineering Journal* 2022, 430, 132618).*

Reply: Thanks for Reviewer 3's valuable comments. Firstly, we searched two literatures Reviewer 3 mentioned in the comment, however, the literature "*Angew. Chem. Int. Ed.* 2020, 59, 1-6" does not exist and is actually a combination of several articles: "Cover Picture: Chameleon Metals: Autonomous Nano-Texturing and Composition Inversion on Liquid Metals Surfaces (*Angew. Chem. Int. Ed.* 2020, 59, 1)", "Inside Back Cover: The Manganese(I)-Catalyzed Asymmetric Transfer Hydrogenation of Ketones: Disclosing the Macrocyclic Privilege (*Angew. Chem. Int. Ed.* 2020, 59, 2)" and an Editorial by Dr. Neville Compton: "The Home of Excellent Chemistry (*Angew. Chem. Int. Ed.* 2020, 59, 4-7)". Hence, we suppose that Reviewer 3 put the wrong literature number here. In that case, we can only compare with the second literature: *Chemical Engineering Journal* 2022, 430, 132618.

It is worth to mention that the literature (*Chemical Engineering Journal* 2022, 430, 132618) mentioned by Reviewer 3 mainly focused on the Pd(II) recovery from high-level radioactive liquid waste (HLLW). However, the focus of our POPcarbene adsorbent is gold recovery from e-waste water and cannot be directly compared with the palladium recovery. Moreover, although similar in the s-triazine structure, the mechanisms raised in the two works are completely different. For the reported literature (*Chemical Engineering Journal* 2022, 430, 132618), it was proved that anion exchange mechanism takes place in the whole process. In contrast, the POPcarbene adsorbent introduce a novel mechanism to recover gold from

e-waste solutions by carbene chemistry (metal-carbene binding affinity, evidenced by XPS and NMR analysis.), which, to the best of our knowledge, have never been investigated. It can be foreseen that such findings can expand "carbene chemistry" further to materials science. Additionally, the uptake of our material--POPcarbene adsorbent reached 2090 mg/g for Au, largely exceeding the one in the literature (428.6 mg/g for Pd).

Comment 2: The morphology modulation of P triaz-CN-A is not fine enough and the resulting specific surface area is low.

Reply: Thanks for Reviewer 3's valuable comments and suggestions. We have improved the preparation process for P triaz-CN-A. Specifically, we extended the time of crosslinking & solvent exchange process from 2 hours to 4 hours, and in the process of supercritical CO₂ drying process, the liquid CO₂ cleaning time was extended to 10 h to completely remove the residual DMF in the P triaz-CN-A. After certain adjustment of preparation methods, the specific surface area of P triaz-CN-A can reach 332 m²/g (**Fig. R3**) and the morphology of modulated P triaz-CN-A is shown in the **Fig. R18**.

Fig. R18 SEM image of modulated P triaz-CN-A.

Revisions made, main text (Page 7 line 133): “Hence, N₂ sorption (at 77 K) measurement was performed to access the pore characteristics of the PtriAZ-CN-A, which based on Brunauer-Emmett-Teller (BET) equation. Its specific BET surface area is calculated to be 332 m² g⁻¹ (**Fig. 1c**), which is fairly acceptable for porous materials prepared free of external templates.”

Revisions made, main text (Page 5 line 97): new **Fig. 1c** was added in main text to replace previous figure.

Fig. R3 N₂ sorption isotherms and pore size distribution (insets) for modified PtriAZ-CN adsorbent after supercritical CO₂ drying. (**Fig. 1c** in revised manuscript)

Comment 3: *As for the gold extraction from CPU, the mechanism of elution regeneration is needed for further investigation. Is there some variation in the microstructure of recycled materials? In addition, the number of cycling recovery is too low and it is recommended to increase to more than 10.*

Reply: Thanks for Reviewer 3’s valuable comments and suggestions. In this study, we used acidic thiourea solution to regenerate Au-loaded PtriAZ-CN-A in desorption process. In a typical desorption experiment, the PtriAZ-CN-A (Au) was desorbed with 20 mL elution solution containing a 1:1 (v/v) solution of thiourea (1 M) and HCl (1 M) under stirring at 60 °C for 8 h.

The adsorbent was then filtered and washed with DI water, and subsequently dried in convection oven at 60 °C for 12 h before being subjected to another adsorption process. For the mechanism of elution regeneration, thiourea ((NH₂)₂CS) used as a gold extracting agent has shown excellent performance. (*Hydrometallurgy* 115-116, 30-51 (2012).). Typically, in acidic conditions, thiourea dissolves gold, forming a cationic complex; the reaction is rapid and gold extraction efficiencies of up to 99 % can be achieved. The anodic reaction follows the equation:

As for the P_{triaz}-CN-A, C5-proton is highly active and a reversible proton exchange take place between H₂O and P_{triaz}-CN-A constantly in aqueous solutions. During the adsorption process, the N-heterocyclic carbene sites can coordinate with gold ions, followed by C-Au bond formation and reductive immobilization of Au³⁺ into Au⁰. When thiourea stripped gold from N-heterocyclic moiety, the N-heterocyclic carbene sites will catch protons in the aqueous solution and recharged, converting back to the original state (**Fig. R19**).

Fig. R19 A proposed mechanism of elution regeneration for P_{triaz}-CN-A (Au).

Moreover, the variations in the microstructure of recycled P_{triaz}-CN-A after 11 cycles were investigated by SEM characterization and the BET surface area was tested by N₂ sorption (at 77 K) measurement. As shown in **Fig. R20**, before adsorption, a great number of particles can be clearly observed as a secondary structure motif, which stems from NH₃-triggered, intramolecularly crosslinked P_{triaz} chains. After the 11th adsorption-

desorption cycles, there existed some variations in the microstructure of recycled P triaz-CN-A, whose particles appear to aggregate and the boundaries between particles also get vague. Furthermore, the BET surface area of P triaz-CN-A was reduced to $53 \text{ m}^2\text{g}^{-1}$. (**Fig. R21**)

Fig. R20 The SEM images of original P triaz-CN-A (a) and recycled P triaz-CN-A.

Fig. R21 N_2 sorption isotherms for recycled P triaz-CN-A after 11 cycles.

Furthermore, the reusability of the material is of vitality for practical applications. Multiple adsorption-desorption experiments were performed to evaluate P triaz-CN-A's performance according to the Reviewer 3's valuable suggestions. **Fig. R17** shows the results of reusability of P triaz-CN-A for Au^{3+} recovery (11 cycles). The results demonstrate that the recovery efficiencies for the P triaz-CN-A were all above 95 % after six cycles, indicating its excellent reusability. The recovery efficiency

decreased continuously in the subsequent cycles owing to the loss of the free spaces of adsorbents due to the continuous enrichment of the reduced Au nanoparticles in the networks of P triaz-CN-A.

Revisions made, main text (Page 13 line 254): new **Fig. 3d** was added in main text to replace previous Figure.

Fig. R17 Regeneration and reusability of P triaz-CN-A. (**Fig. 3d** in revised manuscript).

Revisions made, main text (Page 14 line 290): “**Fig. 3d** shows the results of 11 cycles of the reusability of P triaz-CN-A for Au³⁺ recovery. The results demonstrate that the recovery efficiencies for the P triaz-CN-A were all above 95 % after six cycles, indicating its excellent reusability. The REE decreased continuously in the subsequent cycles owing to the loss of the free spaces of adsorbents due to the continuous enrichment of the reduced Au nanoparticles in the networks of P triaz-CN-A. Furthermore, the regenerated P triaz-CN-A still exhibits more than 60 % elution efficiency (EEE) for Au³⁺ after eleven adsorption-desorption cycles (**Fig. S21**).

Comment 4: Generally speaking, an adsorbent with a K_d value 10^4 mL g^{-1} is regarded to possess excellent adsorption performance. From Figure 3, $K_d(\text{Au})$ is only 998. Evidently, the proposed POPcarbene adsorbent does not possess superior selectivity, unfavorable for the actual application in the complex medium.

Reply: Thanks for Reviewer 3's valuable comments and suggestions. K_d is the distribution coefficient used to compare the affinity of the adsorbent for a certain ion. Particularly, in our previous manuscript, K_d was obtained in a complex environment where many interfering ions coexist and Au^{3+} concentration was as low as 5 ppm, this may result in a lower K_d value. Although the K_d for Au^{3+} is only 998 in the manuscript, the K_d values for other interfering ions (e.g. Cu^{2+} and Ni^{2+} , which are major metal ions in electronic wastewater) are very low, and the K_d values for Cu^{2+} and Ni^{2+} are 0.22 and 0.28, respectively. Moreover, the K_d -based ion selectivity coefficient is ~ 7 for $\text{Au}^{3+}/\text{Pt}^{2+}$, 3.6×10^3 for $\text{Au}^{3+}/\text{Ni}^{2+}$, 4.5×10^3 for $\text{Au}^{3+}/\text{Cu}^{2+}$, 4.5×10^3 for $\text{Au}^{3+}/\text{Co}^{2+}$, 1.2×10^4 for $\text{Au}^{3+}/\text{Zn}^{2+}$, and close to infinity for $\text{Au}^{3+}/\text{Mg}^{2+}$. This data proves the preferential uptake of P triaz-CN-A to noble metal ions (Au^{3+} and Pt^{2+}) over non-precious metals in the e-waste. Based on practical conditions, we suppose that the preferential selectivity of Au^{3+} over other metal ions, e.g. Au^{3+} over interfering ions (the ratio of K_d values of the according metal ions), can be evaluated more practically in this way.

Furthermore, to simulate more realistic electronic wastewater, the prepared e-waste solution contains multiple interfering ions including Co^{2+} , Zn^{2+} , Cu^{2+} , Ni^{2+} , Mg^{2+} , and Cr^{2+} with concentrations at 300 ppm, and Au^{3+} and Pt^{2+} with a concentration of 5 ppm. The affinity of P triaz-CN-A for Au^{3+} is quantified by comparing the distribution coefficients (K_d , mL/g) of different ions. As can be seen from **Table R2**, P triaz-CN-A exhibits the largest distribution coefficient for Au^{3+} , and the K_d -based ion selectivity coefficient is ~ 6 for $\text{Au}^{3+}/\text{Pt}^{2+}$, 2.2×10^3 for $\text{Au}^{3+}/\text{Ni}^{2+}$, 1.1×10^4 for $\text{Au}^{3+}/\text{Cu}^{2+}$, 2.8×10^4 for $\text{Au}^{3+}/\text{Co}^{2+}$, 1.1×10^4 for $\text{Au}^{3+}/\text{Zn}^{2+}$, 0.6×10^3 for $\text{Au}^{3+}/\text{Cr}^{2+}$, and 1.8×10^3 for $\text{Au}^{3+}/\text{Mg}^{2+}$. This data proves the preferential uptake of P triaz-CN-A to noble metal ions (Au^{3+} and Pt^{2+}) over non-precious metals in the e-waste solutions.

Revisions made, main text (Page 12 line 245): “As can be seen from **Table S3**, Ptriaz-CN-A has the largest distribution coefficient for Au³⁺, and the K_d-based ion selectivity coefficient is ~6 for Au³⁺/Pt²⁺, 2.2×10³ for Au³⁺/Ni²⁺, 1.1×10⁴ for Au³⁺/Cu²⁺, 2.8×10⁴ for Au³⁺/Co²⁺, 1.1×10⁴ for Au³⁺/Zn²⁺, 0.6×10³ for Au³⁺/Cr²⁺, and 1.8×10³ for Au³⁺/Mg²⁺. This data proves the preferential uptake of Ptriaz-CN-A to noble metal ions (Au³⁺ and Pt²⁺) over non-precious metals in the e-waste solutions.”

Revisions made, supplementary information (Page 25 line 293): new **Table. S3** was added in supplementary information.

Table. R2. The K_d values of multiple metal ions in simulated e-waste solution. (**Table S3** in revised supplementary information)

	Coexisting Ions							
K_d	Au³⁺	Pt ²⁺	Cu ²⁺	Ni ²⁺	Zn ²⁺	Mg ²⁺	Co ²⁺	Cr ²⁺
	1011.582	174.772	0.0948	0.4564	0.0882	0.548	0.0357	1.712

Comment 5: *More characterization about SEM, TEM, HRTEM and XRD of the recovered gold particles are suggested to provide to facilitate readers' understanding.*

Reply: Thanks for Reviewer 3’s valuable comments and suggestions. In order to study the property of recovered gold nanoparticles after the extraction from e-waste solution, the Ptriaz-CN-A (Au) was calcined in air at 900 °C, and the obtained powder was subsequently immersed in concentrated hydrochloric acid (*aq.*). Despite high concentrations of competing ions in e-waste solution (*e.g.* Cu²⁺, Ni²⁺), these competitive ions are easily dissolved in the concentrated hydrochloric acid after soaking. After that, the treated powders were subsequently washed by deionized water and dried in the oven. Obviously, the resulting material reveals pure solid gold particle signals in the PXRD spectrum (**Fig. R22**), the XRD patterns of resulting material displayed distinct peaks that fit well with the metallic gold.

As for the HRTEM characterization of recovered gold particles

suggested by Reviewer 3, since most of the recovered gold particles are stacked layer by layer and aggregated into large nanoparticles, it is not possible to analyze its lattice and diffraction by HRTEM characterization. Based on above analysis, we supplemented the STEM and elemental mapping characterization (**Fig. R23**) to verify the purity of recovered gold particles. Additionally, more characterization about SEM and TEM of the recovered gold particles are shown in **Fig. R24** and **Fig. R25**.

Revisions made, main text (Page 14 line 276): “In order to study the property of recovered gold nanoparticles after the extraction from e-waste solution, the Ptiaz-CN-A (Au) was calcined in air at 900 °C, and the obtained powder was subsequently immersed in concentrated hydrochloric acid (aq.). Despite high concentrations of competing ions in e-waste solution (e.g. Cu^{2+} , Ni^{2+}), these competitive ions are easily dissolved in the concentrated hydrochloric acid after soaking. After that, the treated powders were subsequently washed by deionized water and dried in the oven. Obviously, the resulting material reveals pure solid gold particle signals in the PXRD spectrum (**Fig. S17**), the XRD patterns of resulting material displayed distinct peaks that fit well with the metallic gold. Furthermore, more characterizations about SEM, TEM, STEM and elemental mapping of the recovered gold particles were also performed (**Fig. S18** to **Fig. S20**).

Revisions made, supplementary information (Page 15 line 239): new **Fig. S17** was added in supplementary information.

Fig. R22 XRD pattern of Au nanoparticles that was purified from Ptri az-CN-A. The pattern indicates that Au³⁺ is reduced to Au⁰ after treatment with the Ptri az-CN-A. (**Fig. S17** in revised supplementary information).

Revisions made, supplementary information (Page 15 line 242): new **Fig. S18** was added in supplementary information.

Fig. R23 STEM image and elemental mapping for recovered gold particles. (**Fig. S18** in revised supplementary information).

Revisions made, supplementary information (Page 16 line 244): new **Fig. S19** was added in supplementary information.

Fig. R24 SEM images of recovered gold particles. (**Fig. S19** in revised supplementary information).

Revisions made, supplementary information (Page 16 line 246): new **Fig. S20** was added in supplementary information.

Fig. R25 TEM images of recovered gold particles. (Fig. S20 in revised supplementary information).

***Comment 6:** Considering extremely high amount of Cu over Au, the energy changes of Cu need to be added in DFT calculations.*

Reply: We thank Reviewer 3's valuable comments and suggestions. We supplemented DFT calculations to simulate adsorption and interactions between P_{triaz}-CN-A model unit and CuCl₂, and the corresponding energy difference (ΔE) in the reaction process is shown in **Fig. R26**. In detail, the 1,2,4-triazolium units firstly adsorb neighboring CuCl₂ from the aqueous solution, and the DFT calculated energy difference (ΔE_1) is -0.54 eV (step 1). Then two HCl molecules are removed from acid-base neutralization between the protons in C5 position of 1,2,4-triazolium units with Cl⁻ in CuCl₂, yet the energy difference (ΔE) in the reaction process is positive (2.45 eV) during the removal of two Cl⁻ and the formation of metal-carbene bond. This result indicates the thermodynamical infeasibility of the Cu-carbene bond formation between 1,2,4-triazolium and copper ion at room temperature in this specific case.

Revisions made, main text (Page 19 line 381): “And the positive energy difference was also observed for CuCl₂ adsorption in the reaction process of Cu-carbene bond formation (**Fig. S35**).”

Revisions made, supplementary information (Page 22 line 286): new **Fig. S35** was added in supplementary information.

Fig. R26 DFT calculated energy differences in adsorption process of CuCl₂ on a model unit of PtriAZ-CN-A. (**Fig. S35** in revised supplementary information).

Comment 7: The saturated adsorption capacity of this manuscript is 2.09 g/g. If the author carefully read more recent literatures, it can be found that adsorption performance of the proposed POPcarbene adsorbent is not superior in the field of gold recovery.

Reply: Thanks for Reviewer 3’s valuable comments and suggestions. We have summarized high-quality literatures of powder adsorbents for gold recovery from electronic wastewater in the past three years. Indeed, there are few porous polymer materials with excellent adsorption performance in the field of gold recovery. Although our PtriAZ-CN-A is not superior in this field, the PtriAZ-CN-A has excellent adsorption performance among many powder adsorbents (**Table. R3**).

Revisions made, supplementary information (Page 23 line 289): new **Table. S1** was added in supplementary information to replace previous **Table S2**.

Table. R3. Comparison of maximum adsorption capacity for Au³⁺. (**Table. S1** in revised supplementary information)

Ranking	Material	Maximum adsorption Capacity (mg g⁻¹)	Reference
1	TpTsc COF	4400.23	Chem. Eng. J. 46, 131865 (2021)
2	PAF-1	2629.87	ACS Appl. Mater. Interfaces 12, 30474-30482 (2020).
3	Tp-BTD-AA	3094.6 (Note: under irradiation with 460 nm)	ACS Sustain. Chem. Eng. 10(30), 9719-9731 (2022).
4	Ptriaz-CN-A	2090	This work
5	FMOF-Co	1953.7	Chem. Eng. J. 410, 128360 (2021)
6	iPAF	1927.3	ACS Appl. Mater. Interfaces 14, 25601-25608 (2022).
7	COP-108	1620	Proc. Natl. Acad. Sci. U.S.A. 117, 16174-16180 (2020).
8	TRF	1432	J. Hazard. Mater. 397, 122812 (2020)
9	COP-224	1354	Chem. Mater. 32, 5343-5349 (2020).
10	JNU-1	1124	Angew. Int. Ed. 59, 17607-17613 (2020).
11	Pc-POSS-POP	862.07	ACS Appl. Mater. Interfaces 14, 11803-11812 (2022)
12	V-PPOP-Br	792.22	J. Hazard. Mater. 426, 128073 (2022)
13	UiO-66-BTU	680.2	Chem. Eng. J. 425, 130588 (2021).
14	APC	576	Chem. Eng. J. 438, 135618 (2022).

Comment 8: The preparation of P-CN-A powder material is rather complicated and expensive, lack of superiority in the engineering application.

Reply: Thanks for Reviewer 3's valuable comments. As we mentioned in our manuscript, the preparation of P-CN-A include three steps in polymer synthesis (monomer synthesis, polymerization and anion exchange) and two steps in the posttreatment (TIP process and ammonia treatment/solvent exchange), followed by freeze drying (**Fig. R27**). All procedures are conducted under mild conditions (temperature ≤ 70 ° C, under atmospheric pressure). In comparison, porous polymer materials with high gold uptakes mostly require harsh conditions during synthesis (solvothermal methods, ≥ 100 °C for several days, *e.g.* certain COFs), or complicated synthesis and purification steps (monomer modifications, column chromatography or re-crystallization, *e.g.* certain COFs and PAFs). Nevertheless, the preparation of P-CN-A is easy-to-handle and do not need sophisticated synthesis skills. Also, these adsorbents are highly repeatable in fabrication. In general, we believe the preparation of P-CN-A is a decent method for adsorbent fabrication and can be used for high-capacity gold uptake.

We also did a thorough life cycle assessment and cost analysis for the production of P-CN-A. And please see the **Additional Supplementary Information (Life Cycle Assessment and Cost Analysis)** for detailed data and analysis. Furthermore, our data collection and analysis are considered on a laboratory scale. As shown in **Fig. R7**, the results indicate that there is much room to reduce the environmental impact and cost of P-CN-A production when they are produced at scale, and show emerging potentials for industrial productions.

Editorial note: Redacted images in the final step represented "freeze-drying" (left) or "supercritical CO₂ drying" (right)

Figure R27 Fabrication process for Ptriiaz-CN-A.

Revisions made, main text (Page 20 line 401): new Fig. 6a and Fig.6b was added in main text.

Fig. R7 (a) Relative environmental impacts of 1 g Ptriiaz-CN-A production at different preparation scales. The higher environmental impact in each category is normalized. (b) Production cost distribution of raw materials, electricity in each step for the synthesis of 1 g Ptriiaz-CN-A at different preparation scales. (**Scale-1**: small dose feeding, data based on our current research; **Scale-2**: big dose feeding, data based on the maximum production scale of the laboratory.) (Fig. 6 in revised manuscript)

Revisions made, main text (Page 20 line 408): “**Life cycle assessment (LCA) and cost analysis.** LCA is a systematic tool for determining the environmental impact of a product or a process across its entire life cycle or a portion of its life cycle. So, the environmental impacts of PtriAZ-CN-A production prepared at different scales in this research were analyzed using the “cradle-to-gate” LCA approach (The detailed data and analysis are summarized in **Additional Supplementary Information**-Life Cycle Assessment and Cost Analysis). **Fig. AS1** shows the system boundary for this LCA study and the whole data was at the lab-scale. As shown in **Fig. 6a**, it is clear that the PtriAZ-CN-A prepared at Scale-2 has much lower environmental impacts across all categories. This is due to the reason that equipment can be operated under full load conditions and resources can be maximized. This indicates that there is much room to reduce the environmental impact of PtriAZ-CN-A production when they are produced on an industrial scale.

Cost analysis of raw materials and electricity in PtriAZ-CN-A preparation process in two different scales are presented in **Fig. 6b** (The detailed data and analysis are summarized in Additional Supplementary Information (Life Cycle Assessment and Cost Analysis)). The production cost for synthesis of 1 g PtriAZ-CN-A in Scale-1 and Scale-2 is around 117.0 CNY and 107.3 CNY, respectively, and more than 95 % of which are spent on raw materials. Although this production cost seems high, it should be noted that the cost here is based on our laboratory data. It is clear that the cost preparation is reduced when we scale up production under laboratory conditions (Scale-2). Therefore, the preparation cost of this adsorbent can be greatly reduced. Furthermore, the value of the gold captured by 1 g of PtriAZ-CN-A was approximately 795.2 CNY (gold price: about 380.5 CNY/g), and the profit margin will continue to increase by regenerating PtriAZ-CN-A. Therefore, PtriAZ-CN-A contributes a green and sustainable method for gold extraction from e-waste solution.

Reviewer 4

In this study, it has been studied on the a poly(ionic liquid)-derived porous organic polycarbene (POPcarbene) adsorbent with superior gold-capturing capability. It has been also utilized theoretical calculations by Density Functional Theory. Study and its results are interesting and publishable after the authors should address the following comments.

***Comment 1:** Line 59 on page: Authors stated the reference of 15 as “Yavuz et al”, however, that reference is not a reference by “Yavuz et al”, so all references should be checked and corrected.*

Reply: We thank Reviewer 4’s valuable comments and suggestions. After checking the literatures, we found that the corresponding author of the reference 15 is Yavuz. However, owing to the citing format, we need to mention the first author here as “Hong et al.”, so we have revised the corresponding descriptions in the manuscript accordingly.

***Revisions made, main text (Page 3 line 65):** “Hong et al.¹⁵ developed a family of porphyrin-phenazine-based polymers...”*

***Comment 2:** Last part of the Introduction section has some sentences about results/discussion/conclusion; this part should only have the aims of the study, so it should be corrected.*

Reply: We thank Reviewer 4’s valuable comments. We have revised the last part of the Introduction section in the manuscript accordingly.

***Revisions made, main text (Page 4 line 83):** “In this study, we report a stable, easy-to-construct porous organic polycarbene (POPcarbene) network for high-performance gold extraction from its waste electronic materials. By taking advantage of an ammonia-catalyzed molecular crosslinking mechanism, the poly(1,2,4-triazolium)s as polycarbene precursor can be processed into a covalently locked porous polymer. The as-synthesized POPcarbene adsorbent are aiming for gold captures in an efficient and selective way, even for gold in a complex multi-ionic solution*

containing Au³⁺, Pt²⁺, Cu²⁺, Mg²⁺, Ca²⁺, Zn²⁺, Co²⁺, and Ni²⁺. The ppb-level of gold ion uptake in its aqueous solution are also considered, judging whether the adsorbent can still enrich a trace amount of gold or not. We also foresee that the mechanism of “gold metallurgy” can be revealed by proving the formation of Au-carbene bonds *via* X-ray photoelectron spectroscopy (XPS) analysis and nuclear magnetic resonance (NMR) spectroscopy. Moreover, density functional theory (DFT) calculations can strengthen our understanding for the gold extraction selectivity and mechanisms. Ultimately, the production of this POPcarbene adsorbent will be subjected to a LCA and cost analysis to determine their environmental impact and cost when producing such adsorbents at scale.”

Comment 3: Some references should be given for the sentence located at line 127 on page 6.

Reply: We thank Reviewer 4’s valuable suggestions. We have cited two references in the positions Reviewer 4 mentioned in the revised manuscript (Page 7 line 137) (*Carbon*. 169, 205-213 (2020) and *Langmuir*. 33, 11138-11145 (2017)).

Revisions made, main text, References (Page 26 line 546):

[27] Jagiello, J., Kyotani, T. & Nishihara, H. Development of a simple NLDFT model for the analysis of adsorption isotherms on zeolite templated carbon (ZTC). *Carbon*. 169, 205-213 (2020).

[28] Kupgan, G., Liyana-Arachchi, T. P. & Colina, C. M. NLDFT pore size distribution in amorphous microporous materials. *Langmuir*. **33**, 11138-11145 (2017).

Comment 4: Bigger geometries for the mechanism steps stated on Figure 5 should be given in supporting information.

Reply: We thank Reviewer 4’s valuable comments and suggestions. We have enlarged the geometries for the mechanism steps stated on **Fig. 5** and placed them in the revised supplementary information accordingly.

Revisions made, supplementary information (Page 19 line 268): new **Fig. S28** was added in supplementary information.

Fig. R28 Proposed mechanism of Au^{3+} adsorption-reduction on the Ptriaz-CN-A (bigger geometries). (Fig. S28 in revised supplementary information).

Revisions made, supplementary information (Page 20 line 270): new Fig. S29 was added in supplementary information

Fig. R29 Adsorption energy (E_{ads}) in step-1 and binding energy (E_{bind}) in sequential step-2 to step-4 of AuCl_4^- and PtCl_4^{2-} with a model unit of Ptriaz-CN-A (bigger geometries). (Fig. S29 in revised supplementary information).

Revisions made, supplementary information (Page 20 line 272): new Fig. S30 was added in supplementary information

Fig. R30 DFT calculated energy differences in adsorption process of AuCl₄⁻ and PtCl₄²⁻ on a model unit of Ptri az-CN-A (bigger geometries). (Fig. S30 in revised supplementary information).

Comment 5: Authors stated at line 30 on page 17 that “... in spite of the negligible energy barrier in step 2.” However, no further information about on the transition state calculations, details for the TS calculations, geometry, activation barrier value, characterization of the TS geometry etc., These should be stated in the manuscript.

Reply: Thanks for the Reviewer 4’s valuable comments and suggestions. We apologize for the misunderstanding raised here, the mentioned “negligible energy barrier” refers specifically to the energy difference (ΔE) in reaction process. And we have corrected it into “energy difference” in the revised manuscript. But according to your constructive suggestions, we have searched transition state (TS) to the best of our ability in a relatively simple way. Considering the limited time, we used ORCA software to search TS structures from the geometry of reactants and products by Nudged Elastic Band with TS optimization (NEB-TS) analysis rather than

conventional searching methods in Gaussian 09. Specifically, the basis sets and functionals we used were in consistent with them used in Gaussian 09 (TPSSH/def2-svp), and D3 keywords were added to consider the dispersion effect and perform dispersion correction. Then we optimized the structure of the speculated “reactants” and “products”, and performed NEB calculation tasks to obtain the minimum energy path (MEP) of the reaction. Next, we extracted the structure ST1 with the highest energy point in the MEP to be the TS structure. After that, the obtained TS geometry was optimized by Gaussian and calculated to get the single point energy. Finally, we used the difference between the electron energy of the TS structure and the electron energy of the reactant structure as the energy barrier during the reaction process. The relative MEP figures and energy profiles of reaction process for both PtCl_4^{2-} and AuCl_4^- are shown in **Fig. R31** to **Fig. R34**. Typically, for the reaction process of AuCl_4^- , the energy of the TS is a little lower than that of reactant, and this may be a negative TS. This means that the reaction process of AuCl_4^- from Step-1 to Step-2 may not have TS. By referring to other references, this phenomenon is common in theoretical calculations. (*Phys. Chem. Chem. Phys.* **12**, 3984-3997 (2010)., *J. Phys. Chem. C* **114**, 12271-12279 (2010). and *Angew. Chem. Int. Ed.* **47**, 1946-1950 (2008).)

Fig. R31 Minimum Energy Path figure for PtCl_4^{2-} adsorption from Step-1 to Step-2.

Fig. R32 Energy profiles of the reaction path for PtCl_4^{2-} adsorption from Step-1 to Step-2. The relative energies (ΔE) are noted.

Fig. R33 Minimum Energy Path figure for AuCl_4^- adsorption from Step-1 to Step-2.

Fig. R34 Energy profiles of the reaction path for AuCl_4^- adsorption from Step-1 to Step-2. The relative energies (ΔE) are noted.

Revisions made, main text (Page 19 line 380): “...in spite of the negligible energy difference in step 2.”

Comment 6: What are the spin multiplicity values for DFT calculations in Gaussian? If the spin multiplicity values greater than 1 (singlet) α and β molecular orbitals should be must be taken into attention.

Reply: Thanks for Reviewer 4’s valuable suggestions. Since AuCl_4^- and H_3O^+ have closed shell structures, the spin multiplicity of the overall structure is equal to that of the P-CN-A model unit. In the current work, we used Gaussian 09 software to perform DFT calculation for P-CN-A model unit using functional and basis set of TPSSH/def2-SVP. The detailed spin multiplicity for the P-CN-A unit and the corresponding total energy are shown in the **Table R4**. It is clearly that when the spin multiplicity is 1, P-CN-A model unit has the lowest energy, indicating a closed shell structure for all electrons in the unit model.

Table R4. The spin multiplicity and the total energy of optimized P-CN-A model unit.

Spin multiplicity	Total energy (a.u.)
1	-1593.156976
3	-1593.035539
5	-1592.927759

For the adsorption of AuCl_4^- , in Step-1 and Step-2, the number of electrons in the whole system is even and it is a closed shell system, so the spin multiplicity is 1; in Step-3 and Step-4, the whole system has an odd number of electrons and is an open-shell system, so the spin multiplicity is 2. For the adsorption of PtCl_4^{2-} , the system has an even number of electrons throughout the adsorption process, so it is a closed shell structure with a spin multiplicity of 1.

Comment 7: Some error, called as Spin Contamination $\langle S^2 \rangle$, may be introduced into the calculations where spin multiplicity is utilized. The spin contamination value must be negligible (less than 10%, David C. Young,

Computational Chemistry, 2001 John Wiley & Sons, Inc. page 228). Thus, related with (1), $\langle S^2 \rangle$ values should be given in text.

Reply: We thank Reviewer 4's valuable comments.

- (1) When spin multiplicity = 1, there is no spin contamination, so no S^2 can be found.
- (2) When spin multiplicity = 2, $S^2 = 0.7585$ and 0.7501 before and after annihilation, respectively.

Comment 8: Spin Density values might be given for the atoms (especially gold atoms) on the structure. These values tell us where unpaired electrons are located in the system.

Reply: We thank Reviewer 4's valuable comments. When the spin multiplicity is 1 (Step-1 and Step-2), the numbers of α and β electrons are the same, and therefore the total spin density is 0. For Step-3, the spin multiplicity for system is 2, there will be an unpaired electron in the system. As shown in the **Fig. R35**, the unpaired electron is mainly concentrated on the triazine ring, but not around the Au atom. This may be ascribed to the adsorption-reduction process and the formation of metal-carbene bond for capturing Au ions. In Step-1, the P-triaz-CN-A model unit interacts with free AuCl_4^- , and the valence of Au is +3 at this time. From Step-1 to Step-2, corresponding to the process of deprotonation and coordination with metal ions to form C-Au bond, the reductive elimination between 1,2,4-triazolium and AuCl_4^- takes place and stepwise reduces AuCl_4^- to AuCl_2^- by releasing HCl molecules, and the valence of Au is +1 at this time. Furthermore, the AuCl_2^- continues to be reduced by releasing HCl molecules, and finally becomes stable Au atom state (the valence of Au is 0 at this time), which is coordinated with carbene site. The unpaired electron is mainly concentrating on the triazine ring due to the coupling of σ - π orbitals between Au and the triazine ring, and the electrons are more stable on the π orbital above and below the triazine ring.

Fig. R35 Spin density diagram for Ptiaz-CN-A-Au model unit in Step-3.

***Comment 9:** Some other critical values such as chemical hardness, chemical potential, electronegativity may be calculated and used to comparison for activities These values can be easily calculated by using HOMO and/or LUMO values based on the approximation of Koopmans.*

Reply: Thanks for Reviewer 4's valuable comments.

- (1) When the spin multiplicity is 1, according to the Koopmans theory, hardness = fundamental gap \approx HOMO-LUMO gap, VIP \approx -E(HOMO), VEA \approx -E(LUMO), Mulliken electronegativity \approx -1/2 (VIP+VEA), chemical potential \approx 1/2 (VIP+VEA).
- (2) When the spin multiplicity is 2, conceptual density functional theory (CDFT), we can directly calculate the wave function and energy of N, N+1, N-1 electronic states, and then obtain the parameters.

Revisions made, main text (Page 18 line 371): "Furthermore, some critical values (chemical hardness, chemical potential...), molecular electrostatic potential diagrams and HOMO/LUMO representations in adsorption process were also calculated by Multiwfn software (Table S5, Table S6, Fig. S31 to Fig. S34)."

Revisions made, supplementary information (Page 25 line 295): new **Table S5** was added in supplementary information.

Table R5 Critical values for AuCl₄⁻ adsorption process in DFT calculations. (**Table S5** in revised supplementary information)

Au	System Charge	Spin Multiplicity	HOMO (a.u.)	LUMO (a.u.)	HOMO-LUMO gap (kcal/mol)	Hardness (kcal/mol)	Chemical potential	Electron-negativity
Step 1	2	1	-0.25949	-0.15641	64.684	64.684	-130.491	130.491
Step 2	0	1	-0.13501	-0.08743	29.857	29.857	-69.792	69.792
Step 3	1	2				105.778	-111.837	111.837
Step 4	0	2				82.787	-81.425	81.425

Revisions made, supplementary information (Page 26 line 297): new **Table S6** was added in supplementary information.

Table R6 Critical values for PtCl₄²⁻ adsorption process in DFT calculations. (**Table S6** in revised supplementary information).

Pt	System Charge	Spin Multiplicity	HOMO (a.u.)	LUMO (a.u.)	HOMO-LUMO gap (kcal/mol)	Hardness (kcal/mol)	Chemical potential	Electron-negativity
Step 1	1	1	-0.25419	-0.18324	44.522	44.522	-137.246	137.246
Step 2	1	1	-0.28214	-0.19021	57.687	57.687	-148.202	148.202
Step 3	1	1	-0.21067	-0.18662	15.092	15.092	-124.652	124.652
Step 4	-1	1	-0.03108	-0.01902	7.568	7.568	-15.719	15.719

Comment 10: *How did you characterized the geometries obtained by DFT calculations?*

Reply: Thanks for Reviewer 4's valuable comments and suggestions. As for the geometry of P-CN-A model unit, we firstly constructed the original model according to the chemical structure of P-CN-A, which obtained from the results of experiment (**Fig. R36**). The model was then geometrically optimized by TPSSH/def2-SVP (opt+freq) using Gaussian

09, and the obtained structure showed successful convergence to the right energy and force threshold, and no imaginary frequency for the obtained structure in frequency calculation.

Fig. R36 (a) Chemical structures of PtriAZ-CN-A. (b) optimized geometry of PtriAZ-CN-A model unit.

Subsequently, vibrational infrared frequency was calculated, and it was compared with the experimental values characterize the geometries obtained by DFT calculations. As shown in **Fig. R37**, it is clearly that the distinct stretching bands of PtriAZ-CN-A and simulated PtriAZ-CN-A model unit are basically consistent. Based on the above discussion, we can verify that the geometry of PtriAZ-CN-A model unit obtained from DFT calculation is optimized.

Fig. R37 Vibrational infrared frequencies spectrum of simulated PtriAZ-CN-A model unit and PtriAZ-CN-A.

In our research, the adsorption-reduction process and the formation of metal-carbene bond for capturing gold was reckoned. As for the coordination numbers of carbene sites between Au, we established a

coordination model between P-CN-A and Au based on NMR results (**Fig. R38**), in which two carbene sites can stabilize one Au atom. The model was then geometrically optimized by TPSSH/def2-SVP (opt+freq) in Gaussian 09, and output results showed successful convergence of simulation results and no imaginary frequency in frequency calculation (**Fig. R39**).

Fig. R38 ^1H NMR spectra of ionic liquids monomer 4-cyanomethy-1-viny-1,2,4-triazolium bromide (IL-CN) and IL-CN (1/2Au).

Fig. R39 Optimized geometry of P-CN-A (AuCl_2^-) model unit in Step-2.

Comment II: Vibrational Infrared frequencies can be calculated. They can be compared with the experimental values stated on page 7, and they can be used to characterize the geometries. Additionally, some mentions about the factor that will be used to scale the frequency values should be stated in text. (Frequency values should be scaled to reproduce experimental fundamentals, the factor and its reference(s) should be stated in text).

Reply: Thanks for Reviewer 4's valuable comments and suggestions. Vibrational infrared frequencies spectrums of simulated PtriAZ-CN-A model unit obtained by Gaussian 09 after opt-freq optimization and PtriAZ-CN-A are shown in **Fig. R37**. It is clearly that the distinct stretching bands of PtriAZ-CN-A and simulated PtriAZ-CN-A model unit are basically consistent. Moreover, we set the scale factor for vibrational frequencies as 0.967.

Fig. R37 Vibrational infrared frequencies spectrum of simulated PtriAZ-CN-A model unit and PtriAZ-CN-A.

Comment 12: What are the convergence criteria in calculations? Gradients of root-mean-square (rms) displacement, max displacement, rms force, max force and the self-consistent field (SCF) convergence.

Reply: Thanks for Reviewer 4's valuable comments. In DFT calculations, it has four parameters to judge the convergence of geometric optimization. Using default convergence settings (convergence on RMS density matrix=1.00D-08 within 128 cycles, convergence on MAX density matrix=1.00D-06, and convergence on energy=1.00D-06), the four parameters are: (1) gradients of root-mean-square (rms) displacement < 0.00120; (2) max displacement < 0.00180; (3) rms force < 0.00030; (4) max force < 0.00045.

Comment 13: NBO analysis should be utilized on the geometries and charge values should be mentioned by using the experimental findings.

Reply: We thank Reviewer 4's valuable suggestions. Generally, the charge distribution over the atoms determines donor and acceptor pairs including the charge transfer in the molecule. In order to reveal the charge transfer interaction between Au and P_{triaz}-CN-A model unit, we have checked the surface natural charge distributions for the Au atoms (Step-1 to Step-4). Consequently, as shown in the **Table R7**, we found that the charge-density becomes smaller and closer to zero for the Au atoms when linked with N-heterocyclic moiety, revealing that charge transfer from the carbene site to Au atom, leading to the formation of strong Au-carbene chemical bond during structural accommodation. These results verify that the N-heterocyclic moiety acts as an electron-donating group that increases the electron density of Au atoms by forming C-Au bond and can further stabilize gold nanoparticles generated thereafter, which is in consistent with our experimental findings.

Table R7. The surface charge distributions of Au atoms from Step-1 to Step-4 during adsorption.

	Step-1	Step-2	Step-3	Step-4
Natural Charge	0.42582	0.25540	0.15005	0.09535

Comment 14: Some DFT calculations about solvent effect should be utilized and results should be compared with present values.

Reply: We thank Reviewer 4's valuable suggestions. We have applied solvent effect (water) to perform DFT calculations and compare energy difference throughout the whole adsorption process for Au/Pt with present values. The detailed energy values of each model unit and the energy difference of adsorption process before/after adding solvent effect are shown in the **Table R8 to Table R11**. It is clear that there is marginal change for the total energy of model unit when we take solvent effect in consideration, but the trend of energy difference is the same as before. This

result indicates that the thermodynamical infeasibility of the formation of Pt-carbene bond between 1,2,4-triazolium and Pt ion at room temperature, no matter whether or not solvent effect is applied. In comparison, a stepwise exothermic process is observed from the calculated AuCl_4^- adsorption process. So, the results after adding solvent effects are in consistent with our previous conclusions.

Revisions made, main text (Page 19 line 396): “Moreover, no matter whether or not solvent effect is applied in DFT calculations, the trend of the reaction process is the same (Table S6 to Table S10).”

Revisions made, supplementary information (Page 26 line 298): new Table S7 was added in supplementary information.

Table R8. The energy values of each model unit during the whole AuCl_4^- adsorption process. (Table S7 in revised supplementary information).

Model unit	Charge	Spin	E-A.U. (No Solvent)	E-A.U. (Solvent: water)
	-1	1	-1976.82	-1976.89
	0	1	-460.83	-460.84
	1	1	-76.73	-76.89
	0	1	-76.46	-76.47
	3	1	-1594.84	-1595.30
	2	1	-3571.96	-3572.20
	0	1	-2650.69	-2650.77
	1	2	-1730.02	-1730.10
	0	2	-3706.92	-3707.02

Revisions made, supplementary information (Page 27 line 301): new **Table S8** was added in supplementary information.

Table R9. The energy difference of AuCl₄⁻ adsorption process before/after solvent effect. (**Table S8** in revised supplementary information)

	$\Delta E1$ (eV)	$\Delta E2$ (eV)	$\Delta E3$ (eV)	$\Delta E4$ (eV)
No Solvent	-8.16	-10.61	-12.25	-2.18
Solvent Effect	-0.27	-6.80	-4.63	-0.82

Revisions made, supplementary information (Page 27 line 303): new **Table S9** was added in supplementary information.

Table R10. The energy values of each model unit during the whole PtCl₄²⁻ adsorption process. (**Table S9** in revised supplementary information).

Model unit	Charge	Spin	E-A.U. (No Solvent)	E-A.U. (Solvent: water)
	-2	1	-1960.44	-1960.72
	0	1	-460.83	-460.84
	1	1	-76.73	-76.89
	0	1	-76.46	-76.47
	3	1	-1594.84	-1595.30
	1	1	-3555.90	-3556.12
	1	1	-2634.22	-2634.32

Revisions made, supplementary information (Page 27 line 306): new Table S10 was added in supplementary information.

Table R11. The energy difference of PtCl₄²⁻ adsorption process before/after solvent effect. (Table S10 in revised supplementary information).

	$\Delta E1$ (eV)	$\Delta E2$ (eV)
No Solvent	-16.87	0.54
Solvent Effect	-2.72	3.27

Comment 15: Do the energy values include Zero Point Energy (ZPE) corrections? If it does not contain, ZPE should be calculated and inserted into energy values or any comments on ZPE should be inserted into the related text.

Reply: Thanks for the Reviewer 4's comments and suggestions. We have extracted all energy-relevant results (including the ZPE corrections) and these results are listed below.

Step-1 P-CN-A (AuCl₄)

Zero-point correction= 0.651031 (Hartree/Particle)
 Thermal correction to Energy= 0.697513
 Thermal correction to Enthalpy= 0.698457
 Thermal correction to Gibbs Free Energy= 0.562341
 Sum of electronic and zero-point Energies= -3568.986972
 Sum of electronic and thermal Energies= -3568.940490
 Sum of electronic and thermal Enthalpies= -3568.939546
 Sum of electronic and thermal Free Energies= -3569.075662

Step-2 P-CN-A (AuCl₂)

Zero-point correction= 0.623495 (Hartree/Particle)
 Thermal correction to Energy= 0.665380
 Thermal correction to Enthalpy= 0.666324
 Thermal correction to Gibbs Free Energy= 0.544290
 Sum of electronic and zero-point Energies= -2648.065081
 Sum of electronic and thermal Energies= -2648.023196
 Sum of electronic and thermal Enthalpies= -2648.022251
 Sum of electronic and thermal Free Energies= -2648.144286

Step-3 P-CN-A (Au)

Zero-point correction= 0.619172 (Hartree/Particle)
 Thermal correction to Energy= 0.657326
 Thermal correction to Enthalpy= 0.658271
 Thermal correction to Gibbs Free Energy= 0.541886
 Sum of electronic and zero-point Energies= -1727.707522
 Sum of electronic and thermal Energies= -1727.669367
 Sum of electronic and thermal Enthalpies= -1727.668423
 Sum of electronic and thermal Free Energies= -1727.784808

Step-4 P-CN-A (Au- AuCl₄)

Zero-point correction= 0.624148 (Hartree/Particle)
 Thermal correction to Energy= 0.672367
 Thermal correction to Enthalpy= 0.673311
 Thermal correction to Gibbs Free Energy= 0.531697
 Sum of electronic and zero-point Energies= -3703.972252
 Sum of electronic and thermal Energies= -3703.924034
 Sum of electronic and thermal Enthalpies= -3703.923090
 Sum of electronic and thermal Free Energies= -3704.064704

Fig. R40 Energy-relevant results (including the ZPE corrections) for AuCl₄⁻ adsorption process in DFT calculations.

Step-1 Ptri az-CN-A (PtCl₂⁻)		Step-2 Ptri az-CN-A (PtCl₂)	
Zero-point correction=	0.650941 (Hartree/Particle)	Zero-point correction=	0.626148 (Hartree/Particle)
Thermal correction to Energy=	0.696853	Thermal correction to Energy=	0.667346
Thermal correction to Enthalpy=	0.697797	Thermal correction to Enthalpy=	0.668291
Thermal correction to Gibbs Free Energy=	0.565057	Thermal correction to Gibbs Free Energy=	0.547840
Sum of electronic and zero-point Energies=	-3552.940469	Sum of electronic and zero-point Energies=	-2631.591184
Sum of electronic and thermal Energies=	-3552.894557	Sum of electronic and thermal Energies=	-2631.549985
Sum of electronic and thermal Enthalpies=	-3552.893612	Sum of electronic and thermal Enthalpies=	-2631.549041
Sum of electronic and thermal Free Energies=	-3553.026353	Sum of electronic and thermal Free Energies=	-2631.669491
Step-3 Ptri az-CN-A (Pt)		Step-4 Ptri az-CN-A (Pt- PtCl₂⁻)	
Zero-point correction=	0.619938 (Hartree/Particle)	Zero-point correction=	0.624953 (Hartree/Particle)
Thermal correction to Energy=	0.657869	Thermal correction to Energy=	0.671965
Thermal correction to Enthalpy=	0.658814	Thermal correction to Enthalpy=	0.672910
Thermal correction to Gibbs Free Energy=	0.542170	Thermal correction to Gibbs Free Energy=	0.538608
Sum of electronic and zero-point Energies=	-1711.333251	Sum of electronic and zero-point Energies=	-3671.440291
Sum of electronic and thermal Energies=	-1711.295319	Sum of electronic and thermal Energies=	-3671.393279
Sum of electronic and thermal Enthalpies=	-1711.294375	Sum of electronic and thermal Enthalpies=	-3671.392335
Sum of electronic and thermal Free Energies=	-1711.411019	Sum of electronic and thermal Free Energies=	-3671.526636

Fig. R41 Energy-relevant results (including the ZPE corrections) for PtCl₄²⁻ adsorption process in DFT calculations.

Comment 16: Energy values should be included thermal energy corrections.

Reply: Thanks for the Reviewer 4's comments and suggestions. We have extracted all energy-relevant results (including the thermal energy corrections) and these results are listed below.

Step-1 Ptri az-CN-A (AuCl₄⁻)		Step-2 Ptri az-CN-A (AuCl₂)	
Zero-point correction=	0.651031 (Hartree/Particle)	Zero-point correction=	0.623495 (Hartree/Particle)
Thermal correction to Energy=	0.697513	Thermal correction to Energy=	0.665380
Thermal correction to Enthalpy=	0.698457	Thermal correction to Enthalpy=	0.666324
Thermal correction to Gibbs Free Energy=	0.562341	Thermal correction to Gibbs Free Energy=	0.544290
Sum of electronic and zero-point Energies=	-3568.986972	Sum of electronic and zero-point Energies=	-2648.065081
Sum of electronic and thermal Energies=	-3568.940490	Sum of electronic and thermal Energies=	-2648.023196
Sum of electronic and thermal Enthalpies=	-3568.939546	Sum of electronic and thermal Enthalpies=	-2648.022251
Sum of electronic and thermal Free Energies=	-3569.075662	Sum of electronic and thermal Free Energies=	-2648.144286
Step-3 Ptri az-CN-A (Au)		Step-4 Ptri az-CN-A (Au- AuCl₂⁻)	
Zero-point correction=	0.619172 (Hartree/Particle)	Zero-point correction=	0.624148 (Hartree/Particle)
Thermal correction to Energy=	0.657326	Thermal correction to Energy=	0.672367
Thermal correction to Enthalpy=	0.658271	Thermal correction to Enthalpy=	0.673311
Thermal correction to Gibbs Free Energy=	0.541886	Thermal correction to Gibbs Free Energy=	0.531697
Sum of electronic and zero-point Energies=	-1727.707522	Sum of electronic and zero-point Energies=	-3703.972252
Sum of electronic and thermal Energies=	-1727.669367	Sum of electronic and thermal Energies=	-3703.924034
Sum of electronic and thermal Enthalpies=	-1727.668423	Sum of electronic and thermal Enthalpies=	-3703.923090
Sum of electronic and thermal Free Energies=	-1727.784808	Sum of electronic and thermal Free Energies=	-3704.064704

Fig. R40 Energy-relevant results (including the ZPE corrections) for AuCl₄⁻ adsorption process in DFT calculations.

Step-1 Ptriaz-CN-A (PtCl ₄ ²⁻)		Step-2 Ptriaz-CN-A (PtCl ₄ ²⁻)	
Zero-point correction=	0.650941 (Hartree/Particle)	Zero-point correction=	0.626148 (Hartree/Particle)
Thermal correction to Energy=	0.696853	Thermal correction to Energy=	0.667346
Thermal correction to Enthalpy=	0.697797	Thermal correction to Enthalpy=	0.668291
Thermal correction to Gibbs Free Energy=	0.565057	Thermal correction to Gibbs Free Energy=	0.547840
Sum of electronic and zero-point Energies=	-3552.940469	Sum of electronic and zero-point Energies=	-2631.591184
Sum of electronic and thermal Energies=	-3552.894557	Sum of electronic and thermal Energies=	-2631.549985
Sum of electronic and thermal Enthalpies=	-3552.893612	Sum of electronic and thermal Enthalpies=	-2631.549041
Sum of electronic and thermal Free Energies=	-3553.026353	Sum of electronic and thermal Free Energies=	-2631.669491
Step-3 Ptriaz-CN-A (Pt)		Step-4 Ptriaz-CN-A (Pt- PtCl ₄ ²⁻)	
Zero-point correction=	0.619938 (Hartree/Particle)	Zero-point correction=	0.624953 (Hartree/Particle)
Thermal correction to Energy=	0.657869	Thermal correction to Energy=	0.671965
Thermal correction to Enthalpy=	0.658814	Thermal correction to Enthalpy=	0.672910
Thermal correction to Gibbs Free Energy=	0.542170	Thermal correction to Gibbs Free Energy=	0.538608
Sum of electronic and zero-point Energies=	-1711.333251	Sum of electronic and zero-point Energies=	-3671.440291
Sum of electronic and thermal Energies=	-1711.295319	Sum of electronic and thermal Energies=	-3671.393279
Sum of electronic and thermal Enthalpies=	-1711.294375	Sum of electronic and thermal Enthalpies=	-3671.392335
Sum of electronic and thermal Free Energies=	-1711.411019	Sum of electronic and thermal Free Energies=	-3671.526636

Fig. R41 Energy-relevant results (including the ZPE corrections) for PtCl₄²⁻ adsorption process in DFT calculations.

Comment 17: Density-of-states and Partial Density-of-states and electronic configurations can be calculated in order to compare results.

Reply: We thank Reviewer 4's valuable comments and suggestions. For the reaction process of AuCl₄⁻ on Ptriaz-CN-A, it is clear that the orbital density of Au orbits gradually increases at Fermi level from Step-1 to Step-3, and in Step-3 (the valence state of Au is 0 at this time), only the orbit of Au exists at the Fermi level (**Fig. R42**), indicating the interaction between Au and N-heterocyclics in Ptriaz-CN-A.

Fig. R42 Density-of-States and Partial Density-of-States for the reaction process of AuCl₄⁻ on Ptriaz-CN-A from Step-1 to Step-4.

Comment 18: *Molecular Electrostatic Potential diagrams and HOMO and LUMO representations can be obtained and they can be compared with after/before adsorption.*

Reply: We thank Reviewer 4's valuable comments and suggestions. The detailed contours for Molecular Electrostatic Potential diagrams and HOMO and LUMO representations before/after adsorption are shown in the **Fig. R43** to **Fig. R46**.

Revisions made, main text (Page 18 line 371): "Furthermore, some critical values (chemical hardness, chemical potential...), molecular electrostatic potential diagrams and HOMO/LUMO representations in adsorption process were also calculated by Multiwfn software (**Table S5**, **Table S6**, **Fig. S31** to **Fig. S34**)."

Revisions made, supplementary information (Page 21 line 274): new **Fig. S31** was added in supplementary information

Fig. R43 HOMO and LUMO representations and electrostatic potential diagrams of Ptriaz-CN-A model unit and AuCl₄⁻. (**Fig. S31** in revised supplementary information).

Revisions made, supplementary information (Page 21 line 275): new Fig. S32 was added in supplementary information.

Fig. R44 HOMO and LUMO representations and electrostatic potential diagrams in adsorption process of AuCl_4^- on a model unit of P-CN-A. (**Fig. S32** in revised supplementary information).

Revisions made, supplementary information (Page 21 line 280): new Fig. S33 was added in supplementary information.

Fig. R45 HOMO and LUMO representations and electrostatic potential diagrams of P-CN-A model unit and PtCl_4^{2-} . (**Fig. S33** in revised supplementary information).

Revisions made, supplementary information (Page 22 line 283): new Fig. S34 was added in supplementary information.

Fig. R46 HOMO and LUMO representations and electrostatic potential diagrams in adsorption process of PtCl_4^{2-} on a model unit of P-CN-A. (**Fig. S34** in revised supplementary information).

Comment 19: Some comment for BSSE can be inserted into text.

Reply: Thanks for Reviewer 4's valuable suggestions. We have inserted some comments for BSSE into supplementary information to tell readers: when calculating binding energy, E_{AB} , E_A , E_B should be calculated at the same basis set level, otherwise, BSSE should be taken into consideration. In general, the weak interaction energy between molecules A and B cannot be calculated simply through $E_{\text{interaction}} = E_{AB} - E_A - E_B$, because the decrease of E_{AB} energy relative to $E_A + E_B$ comes from two aspects. On the one hand, it is the real interaction energy between molecules A and B. That's what we want; On the other hand, the basic functions of molecules A and B overlap in the complex system, which is equivalent to increasing the basis group of the complex and reducing the energy of E_{AB} (strictly speaking, the premise is that the theoretical method used is based on the variational principle). If this part of contribution is also incorporated into $E_{\text{interaction}}$, the interaction energy is overestimated (that is, the binding energy is actually not as negative as calculated), so it should be removed, which is called the Basis

Set Superposition Error (BSSE). Therefore, the interaction energy of two molecules should be formulated as $E_{\text{interaction}} = E_{\text{AB}} - E_{\text{A}} - E_{\text{B}} + E_{\text{BSSE}}$. For weak interactions, the proportion of E_{BSSE} in $E_{\text{interaction}}$ is often not small, or even exceeds it. If not corrected, the positive and negative signs may not be correct.

Revisions made, supplementary information (Page 7 line 163): some comments for BSSE were inserted into supplementary information.

REVIEWERS' COMMENTS

Reviewer #1 (Remarks to the Author):

The authors have revised the manuscript extensively. A lot of comments by the four reviewers are addressed and the work is considerably better. I recommend publication but I would like the structural identification to be clear with proper chemical equation displayed in one of the figures. The issue is that amidine vs. triazine is not settled, as the ^{13}C -NMR of the crosslinked polymer shows 168 ppm (which falls within the 166-169 ppm assignment for amidines). Ultimately, one wouldn't need perfectly uniform identification of the functional group if the reproducibility and gold uptake is there. But for the sake of clarity, I would expect the authors to scheme chemical reactions with both amidine and triazine possible formation paths drawn.

Reviewer #2 (Remarks to the Author):

The authors significantly worked on the paper and were able to improve the quality of the manuscript. They well addressed all the questions/concerns raised by me and I believe the manuscript can be accepted with its current version.

Reviewer #3 (Remarks to the Author):

The authors have addressed all issues well. The manuscript has been significantly improved, and thus it is recommended to be published as it is.

Reviewer #4 (Remarks to the Author):

manuscript can now accepted,

Reviewer 1:

The authors have revised the manuscript extensively. A lot of comments by the four reviewers are addressed and the work is considerably better. I recommend publication but I would like the structural identification to be clear with proper chemical equation displayed in one of the figures. The issue is that amidine vs. triazine is not settled, as the ^{13}C -NMR of the crosslinked polymer shows 168 ppm (which falls within the 166-169 ppm assignment for amidines). Ultimately, one wouldn't need perfectly uniform identification of the functional group if the reproducibility and gold uptake is there. But for the sake of clarity, I would expect the authors to scheme chemical reactions with both amidine and triazine possible formation paths drawn

Reply: We thank Reviewer 1's valuable comments and suggestions. Conclusively, we believe both *s*-triazine and amidine structures coexist in the P-CN-A. Hence, we revised Figure 1f, Figure 1g, Figure 4f, Figure S23-25 in the manuscript and Supplementary Information, and schemed chemical reactions with both amidine and *s*-triazine structures for P-CN-A formation in Figure S4 in Supplementary Information to clarify Reviewer 1's concern.

Revisions made, main text (Page 5 line 92): Figure 1f was revised.

Figure 1f solid-state ^{13}C NMR spectra of Ptriaz-CN (bottom line) and the as-synthesized Ptriaz-CN-A (top line).

Revisions made, main text (Page 5 line 92): Figure 1g was revised.

Figure 1g N1s XPS analysis of the as-synthesized Ptriaz-CN-A. The coloured shadings represent internal integral area of different characteristic peaks. N4 (purple), Amidine (blue), Triazine (orange), N1 (red) and N2 (brown).

Revisions made, main text (Page 15 line 296): Figure 4f was revised.

Figure 4f High resolution C 1s XPS spectra before and after Au³⁺ adsorption. The colored shadings represent internal integral area of different characteristic peaks. C3 (purple), C6 (pink), Triazine/Amidine (orange), C5 (red) and -CF₃ (brown).

Revisions made, supplementary information: new Fig. S4 was added to scheme chemical reactions with both amidine and *s*-triazine structures for Ptriaz-CN-A formation.

Fig. S4 Chemical reactions with both amidine and triazine structures for Ptriaz-CN-A formation paths.

Revisions made, supplementary information: Fig. S5 was revised.

Fig. S5 General pathway toward mesoporous Pptriaz-CN-A

Revisions made, supplementary information: Fig. S23 was revised.

Fig. S23 The N 1s XPS spectra of Pptriaz-CN-A before and after Au³⁺ adsorption. The colored shadings represent internal integral area of different characteristic peaks. N4 (purple), Amidine (blue), Triazine (orange), N1 (red) and N2 (brown).

Revisions made, supplementary information: Fig. S24 was revised.

Fig. S24 (a) XPS full spectra of Ptiaz-CN-A before and after Pt²⁺ adsorption. (b) High-resolution XPS spectra of Pt 4f after Pt²⁺ adsorption. The colored shadings represent internal integral area of different characteristic peaks. Pt²⁺ (red). (c) The C 1s XPS spectra of Ptiaz-CN-A before and after Pt²⁺ adsorption. The colored shadings represent internal integral area of different characteristic peaks. C3 (purple), C6 (pink), Triazine/Amidine (orange), C5 (red) and -CF₃ (brown). (d) The N 1s XPS spectra of Ptiaz-CN-A before and after Pt²⁺ adsorption. The colored shadings represent internal integral area of different characteristic peaks. N4 (purple), Amidine (blue), Triazine (orange), N1 (red) and N2 (brown)

Revisions made, supplementary information: Fig. S25 was revised.

Fig. S25 (a) XPS full spectra of Ptriz-CN-A before and after Cu²⁺ adsorption. (b) High-resolution XPS spectra of Cu 2p after Cu²⁺ adsorption. The colored shadings represent internal integral area of different characteristic peaks. Cu²⁺ (red) and Cu¹⁺ (purple). (c) The C 1s XPS spectra of Ptriz-CN-A before and after Cu²⁺ adsorption. The colored shadings represent internal integral area of different characteristic peaks. C3 (purple), C6 (pink), Triazine/Amidine (orange), C5 (red) and -CF₃ (brown). (d) The N 1s XPS spectra of Ptriz-CN-A before and after Cu²⁺ adsorption. The colored shadings represent internal integral area of different characteristic peaks. N4 (purple), Amidine (blue), Triazine (orange), N1 (red) and N2 (brown).